# TOWARDS MULTIPLE CHARACTER IMAGE ANIMATION THROUGH ENHANCING IMPLICIT DECOUPLING

Jingyun Xue[1,2,*] Hongfa Wang[2,3,*] Qi Tian[2,*] Yue Ma[2,4], Andong Wang[2], Zhiyuan Zhao[2], Shaobo Min[2],
Wenzhe Zhao[2], Kaihao Zhang[5], Heung-Yeung Shum[3,4], Wei Liu[2], Mengyang Liu[2,†] Wenhan Luo[4,‡]
[1]Shenzhen Campus of Sun Yat-sen University, [2]Tencent Hunyuan, [3]Tsinghua Univerisity
[4]HKUST, [5]Harbin Institute of Technology, Shenzhen
https://multi-animation.github.io/

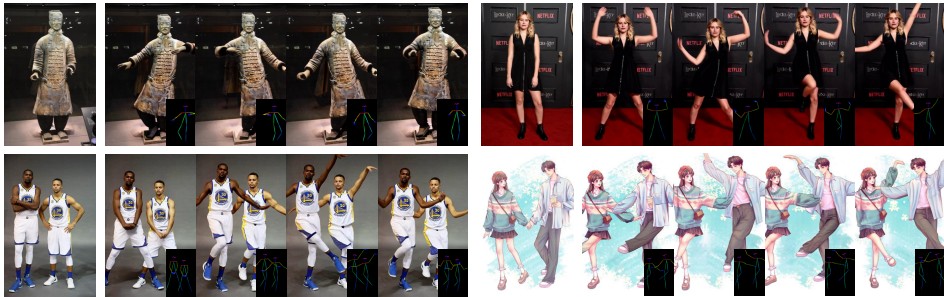

Figure 1: Pose-controllable single-character image animation (top) and dual-character image animation (bottom) given the reference image.

## ABSTRACT

Controllable character image animation has a wide range of applications. Although existing studies have consistently improved performance, challenges persist in the field of character image animation, particularly concerning stability in complex backgrounds and tasks involving multiple characters. To address these challenges, we propose a novel multi-condition guided framework for character image animation, employing several well-designed input modules to enhance the implicit decoupling capability of the model. First, the optical flow guider calculates the background optical flow map as guidance information, which enables the model to implicitly learn to decouple the background motion into background constants and background momentum during training, and generate a stable background by setting zero background momentum during inference. Second, the depth order guider calculates the order map of the characters, which transforms the depth information into the positional information of multiple characters. This facilitates the implicit learning of decoupling different characters, especially in accurately separating the occluded body parts of multiple characters. Third, the reference pose map is input to enhance the ability to decouple character texture and pose information in the reference image. Furthermore, to fill the gap of fair evaluation of multi-character image animation, we propose a new benchmark comprising about $4,000$ frames. Extensive qualitative and quantitative evaluations demonstrate that our method excels in generating high-quality character animations, especially in scenarios of complex backgrounds and multiple characters.

## 1 INTRODUCTION

Character image animation task targets animating a given static character image to a video clip using a sequence of motion signals, such as pose, while preserving the visual appearance. It has attracted much attention and has been extensively explored in research (Zhang et al., 2022). Existing character image animation can be divided into three categories: GAN-based, 3D-based, and

---

*These authors contributed equally to this research
†Project leader
‡Corresponding author

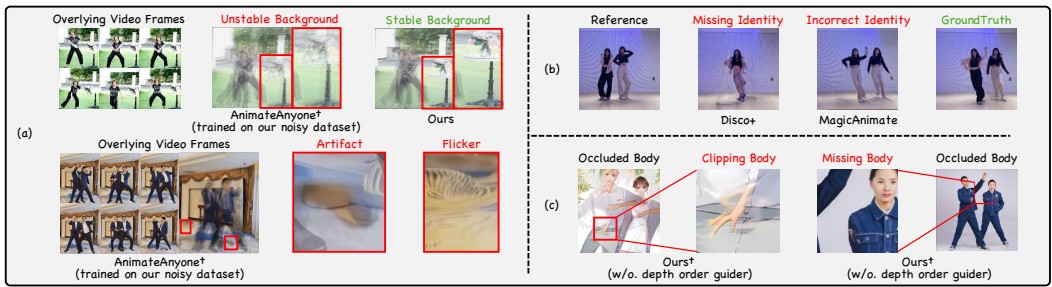

Figure 2: Challenges faced by existing methods: (a) By overlaying video frames, it is observed that models generate unreasonable backgrounds. (b) Models struggle to accurately identify characters in multi-character image animations, resulting in incorrect character identities. (c) Models have difficulty in accurately generating the occluded body parts of multiple characters.

diffusion-based frameworks. GAN-based methods (Siarohin et al., 2019; Wang et al., 2021) leverage a warping function to transfer the reference image into the target pose, and then the GAN model is employed to generate the missing parts. 3D-based methods (Jiang et al., 2023) reconstruct a character avatar from monocular videos and then render the avatar into a character video based on the pose sequence. Diffusion-based methods generate refined character animation by adding conditional control. For example, Hu et al. (2023) generate a series of coherent action videos by incorporating pose sequences, while Xu et al. (2023) do so by integrating semantic map sequences.

Despite generating visually plausible videos, existing methods still face several challenges. GAN-based methods often struggle to generate realistic animations due to their limited ability to transfer motion, which affects details such as hair movement, and facial expressions. 3D-based methods (Jiang et al., 2023; Yu et al., 2023) are generally limited by the precision of SMPL and struggle to derive hand and facial details. In addition, 3D-based methods cannot fill in the empty background left after character movement. Besides, 3D-based methods perform poorly in generating high-quality texture details and clothing movement. In contrast, diffusion-based methods alleviate these problems due to their powerful generative capability. However, diffusion-based methods still face serious challenges when generating higher-quality and more complex character animations. As shown in Fig. 2, existing methods often have the drawback of generating unreasonable backgrounds. Specifically, due to the camera shake and/or video effects popularly present in the background, the model is affected by the noise, leading to abrupt changes, flickering, and artifacts in the background. Additionally, when generating animated videos with multiple characters, existing methods tend to generate chaotic character identities and erroneous occluded body parts.

Our work is dedicated to addressing these challenges. Through our investigation, we have the following findings. First, the background instability results from the model's low robustness to noisy data with background variations in the training set. The common solution is to construct a high-quality training set without background variation, which is costly and limits the size of the dataset, especially considering that data for multiple characters is scarce. We aim for the model to achieve strong generalization with sufficient and cheap training data with background variations. A feasible solution is to decouple the noise features from the video, improving the utility of the noisy data. Second, the errors in multiple character image animation primarily stem from that the model cannot further decouple multiple character features into individual character features. Perhaps we can enable the model to learn this implicit decoupling process through extensive multiple-character data, but a more economical and expeditious approach is to provide guiding information to direct the model in learning this process. These findings indicate that it is extremely important to introduce the implicit decoupling ability into the model for the task of multiple-character image animation.

In this work, to address the aforementioned challenges, we propose a diffusion-based framework for multiple-character image animation, allowing for the input of various types and amounts of guidance information. Based on the mentioned insights, we carefully design three guiders to enhance the implicit decoupling capability of the model. (1) We design an optical flow guider that calculates the background motion optical flow map as guidance. This enables the model to implicitly learn to decouple the background motion into background constants and background momentum during training, and generate a stable background by inputting zero background momentum during inference. (2) We present the depth order guider calculating the order map of the characters, which transforms

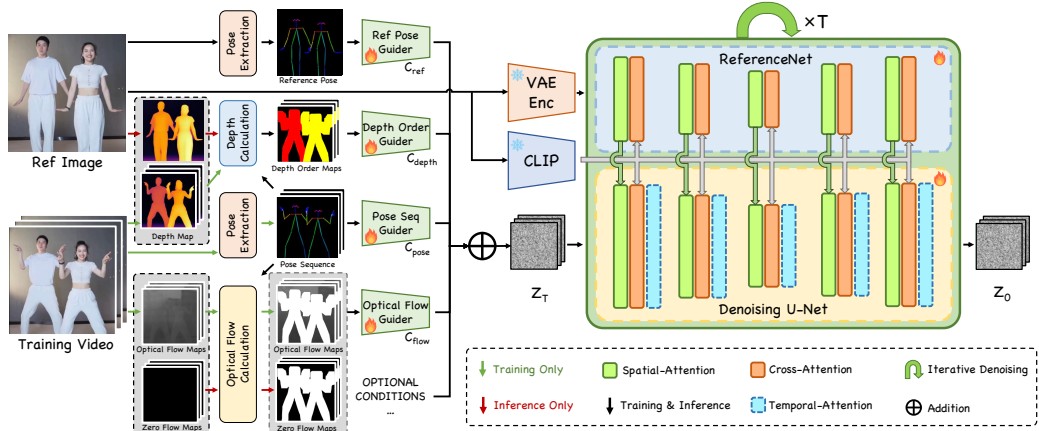

Figure 3: The overview of the proposed framework. The left half illustrates the data flow of the multiple condition guiders, with green and black arrows denoting training data flow, and red and black arrows indicating inference data flow. The gray box represents different inputs for training and inference. The right half shows the denoising U-Net and ReferenceNet.

the depth information into the positional information of multiple characters in terms of their relative front and back positions. This facilitates the implicit learning of decoupling different characters, especially in accurately separating the occluded body parts of multiple characters, leading to more stable multiple-character animation. (3) We introduce the reference pose map to enhance the ability to decouple character texture and pose information in the reference image. Additionally, we propose a multi-character benchmark including about $4,000$ frames, which empowers the community to comprehensively evaluate the generation ability in complex multiple character image animations. To the best of our knowledge, we are the first to collect such a benchmark. We conduct extensive quantitative and qualitative experiments to illustrate the superiority of our approach.

In summary, our contributions are as follows:

- We propose a multi-condition guided framework with multiple guiders for multiple-character image animation, which exhibits strong robustness against noisy data with unstable backgrounds. By leveraging large-scale raw data for training, this framework effectively addresses the multiple-character image animation.

- We enhance the implicit decoupling capability of the model. Technically, the *optical flow guider* decouples the background momentum to ensure background stability, the *depth order guider* provides multiple character positional information to address the occlusion among body parts, and *reference pose guider* inputs the source pose to align the character with the target pose.

- We address the lack of a benchmark in multiple character image animation. A new benchmark called *Multi-Character* Bench is introduced, which contains about $4,000$ frames for fair and comprehensive evaluation.

- Extensive quantitative and qualitative evaluations are conducted using two public datasets and *Multi-Character* Bench. The results show the superiority of the proposed method.

## 2 RELATED WORK

**Diffusion Models for Video Generation.** Diffusion models excel in generating high-quality contents (Esser et al., 2023; Ma et al., 2023; 2024c; Chen et al., 2024b; Feng et al., 2024; Tan et al., 2024; Kong et al., 2025). Ho et al. (2020); Song et al. (2020) propose to generate images using Diffusion models. Many studies (Wu et al., 2023) adapt image synthesis pipelines for video generation. Khachatryan et al. (2023) incorporate motion dynamics in generated frames via cross-frame attention in zero-shot style. Guo et al. (2023b) finetune a plug-and-play motion module that can be integrated into any text-to-image models to obtain animations in a personalized style. Zhang et al. (2024); Chen et al. (2024a) achieve both text-to-video and image-to-video generation of high resolution. Besides, Guo et al. (2023a) extract semantics from images to condition T2V generation.

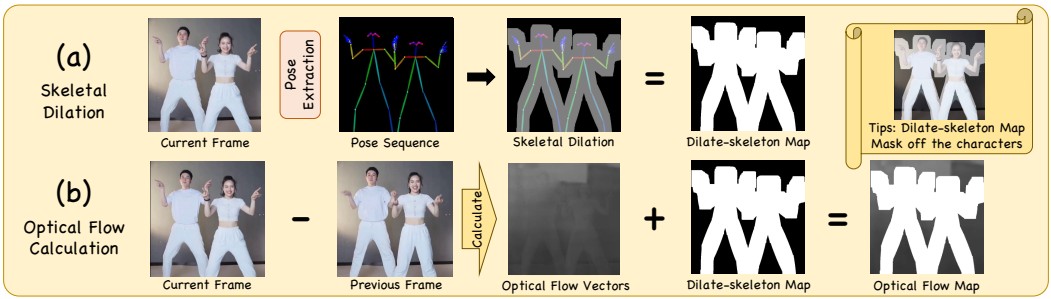

Figure 4: The pipeline for calculating the skeletal dilation map and optical flow map. The optical flow map is obtained by superimposing the skeletal dilation map onto the optical flow vector map.

**Pose-Controllable Character Image Animation.** Generating realistic character video from the driving signal (*e.g.*, key points, semantic maps) has been extensively studied in recent years. Early approaches (Mirza & Osindero, 2014; Wang et al., 2021) employ GAN-based models for conditional video synthesis. Wang et al. (2019) use conditional GANs with optical flow, temporal consistency, and multiple discriminators for diverse pose videos. Chan et al. (2019) transfer movements from a source person to a target person through keypoints. Siarohin et al. (2021) incorporate consistent regions that describe locations, shapes, and poses. To address the pose gap between objects in the source and driving images, Zhao & Zhang (2022) effectively align object poses using thin-plate spline motion estimation and multi-resolution occlusion mask. With the development of the diffusion model, existing methods use pose sequences to guide the character video generation (Zhang et al., 2023). Karras et al. (2023) propose a finetuning framework to adapt the Stable Diffusion model to a pose-and-image guided video synthesis model. Hu et al. (2023) design ReferenceNet to merge detailed human body features, ensuring character appearance consistency, while employing a pose guider and temporal modeling for controllable and continuous movements. To enhance the performance, Wang et al. (2023) disentangle the control conditions (*i.e.*, character foreground, background, and poses) by introducing multiple ControlNets for different feature embeddings and a human attribute pre-training framework is proposed. Moreover, Ma et al. (2024a) propose a two-stage training scheme to address the lack of comprehensive paired video-pose datasets.

## 3 PRELIMINARIES

**Video Latent Diffusion Models.** The powerful text-to-image (T2I) model with the addition of temporal motion modules endows it with the ability to generate video (Guo et al., 2023b), *i.e.*, video latent diffusion model (VLDM). Specifically, inserting a temporal motion module after spatial attention enables the T2I model to generate video. The temporal motion modules run across temporal frames to enhance motion smoothness and content consistency. The encoder of VAE (Kingma & Welling, 2013; Razavi et al., 2019) compresses the video clip $v_{1:n}$ to a latent space feature $z = \mathcal{E}(v_{1:n})$. Besides, the initial input tensor $z \in \mathbb{R}^{c \times h \times w}$ of the T2I model should add a temporal dimension and repeat it $n$ times, becoming $z_{1:n} \in \mathbb{R}^{n \times c \times h \times w}$. VLDM uses U-Net (Ronneberger et al., 2015) or DiT (Peebles & Xie, 2023) to estimate the noise, with the loss function of

$$\mathcal{L} = \mathbb{E}_{\mathcal{E}(v_{1:n}), c, \epsilon_{1:n} \sim \mathcal{N}(0, I), t} \left[ ||\epsilon - \epsilon_\theta(z_{1:n}^t, t, c)||_2^2 \right] , \tag{1}$$

where $\epsilon_\theta(\cdot)$ is the network for predicting noise. $c$ denotes the conditional information. $t$ represents the timestep of denoising process and $z_{1:n}^t$ is the intermediate result of denoising at timestep $t$.

## 4 METHODOLOGY

Given a reference image $I_0$ and pose sequences $\{P_0, P_1, P_2, ..., P_N\}$, we aim to generate character video clip with both plausible motion and faithful visual appearance (foreground and background) regarding the reference image. To address the issues of identity errors and body occlusion in multi-character image animation, or background instability, building large-scale high-quality datasets and conducting extensive training is costly and laborious. In contrast, enhancing the implicit decoupling ability of the model through input guidance is more economical and feasible. To achieve this, we propose a novel framework equipped with multi-condition guiders, as shown in Fig. 3.

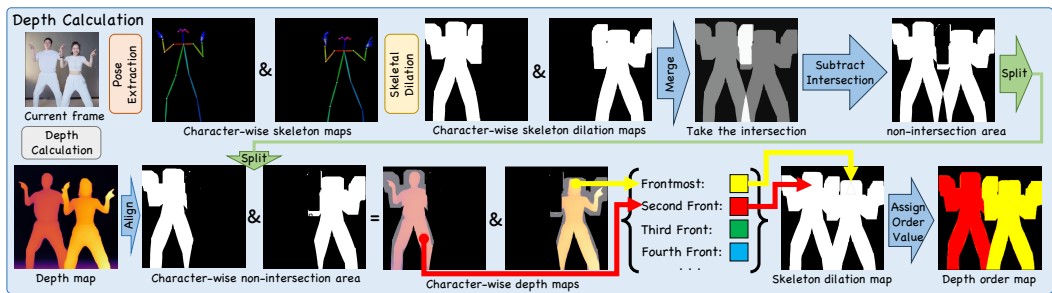

Figure 5: The pipeline for calculating the depth order map, which strictly separates characters and assigns distinct values. The yellow module "Skeletal Dilation" is shown in Fig. 4 (a).

### 4.1 OPTICAL FLOW GUIDER

As shown in Fig. 2 (a), when directly training on noisy data, the model is unable to achieve background alignment due to the interference caused by noise. As a result, there are abrupt changes, flickering, and artifacts in the generated background. We propose using background optical flow maps as guidance to direct the model in learning the implicit decoupling process, which decouples the background into a constant and a momentum. Here, background momentum represents the relative motion of the current frame's background compared to the previous frame, which is equivalent to the optical flow. After decoupling the background motion, the model can learn to generate stable backgrounds even trained on noisy data. During inference, we input zero background motion to achieve stable background. Unlike the work (Liang et al., 2024) using optical flow to enhance temporal consistency, we use the motion vector information in optical flow to decouple noise features.

Fig. 4 presents the pipeline for calculating the background optical flow map in the training stage (Yang et al., 2023; Güler et al., 2018; Sun et al., 2018). The optical flow estimator $\mathcal{E}_{flow}$ (OpenMMLab, 2021) predicts optical flow maps from adjacent frames $\{v_1, v_2..., v_N\}$. The dilation operation is then applied on pose skeletons $\{p_1, p_2..., p_N\}$ to obtain binary mask sequences $\{\mathcal{M}_1, \mathcal{M}_2, \mathcal{M}_3, ..., \mathcal{M}_N\}$, which are used for defining the control region. We blend optical flows with the binary mask and encode them using 8 layers of inflated 3D convolution. The region in blended optical flows is set to $1.0$. In inference, we directly set the background optical flow, *i.e.*, the background momentum, to zero. Formally, the optical flow guider is implemented as:

$$\mathcal{M}_i = \mathcal{D}(p_i), \tag{2}$$

$$c_{flow}^i = g_{\text{op}}\left((1 - \mathcal{M}_i) \odot \mathcal{E}_{flow}(v_i, v_{i-1})\right), \tag{3}$$

where $\mathcal{D}$ represents the dilation operation, $\odot$ is Hadamard product, and $g_{\text{op}}$ is the optical flow guider.

### 4.2 DEPTH ORDER GUIDER

Concurrent methods struggle to accurately identify different characters and generate correct identities. They lack spatial order information about different characters in front of the camera, which is critical for generating occluded body parts. To address this issue, we design a depth order guider that can simultaneously enhance the implicit decoupling of characters into individual ones and provide characters' spatial order information. Specifically, it calculates the depth order map of characters, decoupling multiple characters into individuals while providing their sequential positional/depth relationships in front of the camera. The regions of different characters are separated by assigning distinct values, with occluded body parts being strictly assigned to characters positioned in the front.

Fig. 5 illustrates calculating the depth order map during training. For each frame, we extract individual character poses and calculate their skeletal dilation maps. Subsequently, we merge the character-wise skeleton dilation maps to get the intersection and subtract it to derive the non-intersection area. Next, split by characters, as shown by the green arrow in Fig. 5, we have the character-wise non-intersection area maps. Then, we align them with the depth maps, compute the average depth of the non-intersection region for each character, and sort them by the average depth. We assign values to each character's region based on their position ranking. For example, as shown in Fig. 5, the frontmost character region gets "yellow value" while the second front character region gets "red value".

Table 1: Quantitative comparison on Tiktok dataset. The best and second-best results are indicated in red and blue respectively. AnimateAnyone† is trained on our noisy dataset. Ours* is trained on the Tiktok training set.

| Method | FID↓ | SSIM↑ | PSNR↑ | LPIPS↓ | L1↓ | FID-VID↓ | FVD↓ |
|---|---|---|---|---|---|---|---|
| MRAA (Siarohin et al., 2021) | 54.47 | 0.672 | 29.39 | 0.296 | 3.21E-04 | 66.36 | 284.82 |
| TPSMM (Zhao & Zhang, 2022) | 53.78 | 0.673 | 29.18 | 0.299 | 3.23E-04 | 72.55 | 306.17 |
| DreamPose (Karras et al., 2023) | 79.46 | 0.509 | 28.04 | 0.450 | 6.91E-04 | 80.51 | 551.56 |
| DisCo (Wang et al., 2023) | 51.29 | 0.699 | 28.70 | 0.333 | 1.10E-04 | 61.41 | 379.56 |
| DisCo+ (Wang et al., 2023) | 48.29 | 0.713 | 28.78 | 0.320 | 1.03E-04 | 52.56 | 334.67 |
| MagicAnimate (Xu et al., 2023) | 32.09 | 0.714 | 29.16 | 0.239 | 3.13E-04 | 21.75 | 179.07 |
| MagicPose (Chang et al., 2023) | 25.50 | 0.752 | 29.53 | 0.292 | 0.81E-04 | 46.30 | 216.01 |
| AnimateAnyone (Hu et al., 2023) | - | 0.718 | 29.56 | 0.285 | - | - | 171.90 |
| AnimateAnyone† | 54.42 | 0.685 | 29.01 | 0.316 | 1.06E-04 | 47.93 | 236.28 |
| Ours* | 29.15 | 0.735 | 29.61 | 0.287 | 0.79E-04 | 35.28 | 153.47 |
| Ours | 27.70 | 0.760 | 29.70 | 0.272 | 0.73E-04 | 14.30 | 117.81 |

Table 2: Quantitative comparison on TED-talks dataset. The best and the second-best results are indicated in red and blue respectively. AnimateAnyone† is trained on our noisy dataset.

| Method | FID↓ | SSIM↑ | PSNR↑ | LPIPS↓ | L1↓ | FID-VID↓ | FVD↓ |
|---|---|---|---|---|---|---|---|
| MRAA (Siarohin et al., 2021) | 50.36 | 0.762 | 31.90 | 0.266 | 0.50E-04 | 82.79 | 493.02 |
| TPSMM (Zhao & Zhang, 2022) | 23.71 | 0.771 | 32.30 | 0.252 | 0.49E-04 | 32.12 | 260.67 |
| DisCo (Wang et al., 2023) | 75.48 | 0.575 | 27.99 | 0.309 | 1.21E-04 | 66.18 | 393.04 |
| DisCo+ (Wang et al., 2023) | 63.28 | 0.596 | 28.12 | 0.300 | 1.11E-04 | 55.81 | 343.20 |
| MagicAnimate (Xu et al., 2023) | 41.58 | 0.529 | 28.28 | 0.310 | 1.73E-04 | 33.61 | 223.54 |
| MagicPose (Chang et al., 2023) | 23.39 | 0.723 | 30.08 | 0.236 | 0.81E-04 | 27.53 | 214.23 |
| Moore-AnimateAnyone | 25.93 | 0.710 | 30.99 | 0.310 | 0.46E-04 | 41.20 | 262.49 |
| AnimateAnyone† | 47.68 | 0.691 | 29.59 | 0.283 | 1.15E-04 | 30.07 | 241.76 |
| Ours | 18.21 | 0.779 | 30.88 | 0.198 | 0.46E-04 | 10.24 | 81.73 |

The result is the depth order map. During inference, the calculation of the average depth value of the skeleton dilation areas is skipped. Instead, we directly compute the order values of the characters in the reference image and assign these values to the corresponding areas of the character dilation skeletons. Given a training video $v$, where the $i$-th frame is denoted as $v_i$, supposing that there are $J$ characters with pose skeleton $\{p_1, ..., p_J\}$ in $v_i$. The training processes can be illustrated as

$$a_{i,1}, ... a_{i,J} = \mathcal{D}(p_1, ..., p_J),\tag{4}$$

$$m_{i,j} = a_{i,j} - \left(1 - \bigcap_{k \in \{1,...,J\}} (a_{i,k})\right), j \in \{1..J\},\tag{5}$$

$$rank_j = f_{\text{sort}}(m_{i,1} \odot f_{\text{d}}(v_1), ..., m_{i,j} \odot f_{\text{d}}(v_i)),\tag{6}$$

$$c_{\text{depth},j} = m_{i,j} \odot L_{rank_j} + (1 - m_{i,j} \odot c_{\text{depth},j+1}), c_{\text{depth},J+1} = 0,\tag{7}$$

$$c_{\text{depth}} = g_{\text{dp}}(c_{\text{depth},1}),\tag{8}$$

where $\mathcal{D}$ represents the dilation operation, $a_{i,j}$ denotes the character-wise skeleton dilation map. $\cap$ is intersection operation and $m_{i,j}$ is the non-intersection region of each character. $f_{\text{d}}$ represents the depth calculation network, $f_{\text{sort}}$ is average depth sorting (descending order), and $rank_j$ is the depth ranking of character $j$. $L_{rank_j}$ denotes the value assigned to $j$ based on depth ranking and $c_{\text{depth},j}$ is the depth order map of the farthest $j$ characters. $g_{\text{dp}}$ represents the depth order guider.

## 4.3 REFERENCE POSE GUIDER

The positions of characters in inference images and pose sequences often exhibit inconsistencies. Diffusion-based methods employ spatial cross-attention to facilitate the interaction between the pose frame and reference image, enabling it to learn character texture information from the reference images and accurately map them to the corresponding pose positions. We realize that this process involves the model implicitly decoupling the character's pose and fine-grained texture information from the reference image, and then mapping the texture to the target position by aligning the character's pose with the target pose. Consequently, we can assist the model in this decoupling process by introducing the reference pose as a prior. Since the reference pose map and the pose frame are consistent in format, the model can directly align the pose and map the characters efficiently without

| Reference | MRAA* | TPSMM* | DisCo | DisCo+ | MagicAnimate | MagicDance | Ours | GroundTruth |

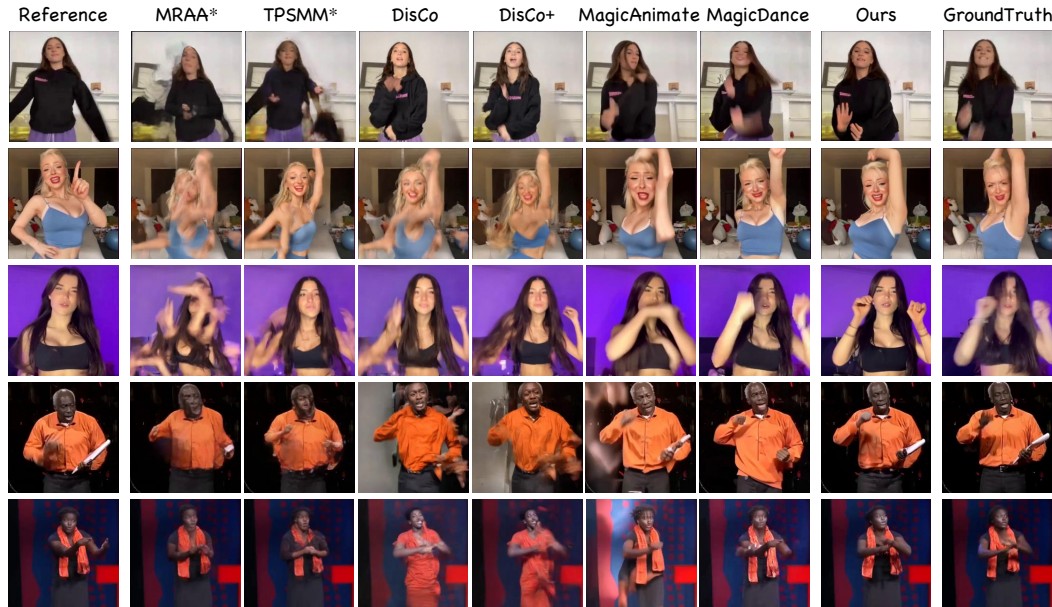

Figure 6: Qualitative comparisons between baselines and our approach on dataset Tiktok (Top three rows) and TED-talks (bottom two rows). MRAA* and TPSMM* present these methods utilizing ground-truth videos as driving signals.

calculating poses from the reference image. In this way, the model can reduce the learning load and focus on mapping character textures. Given the reference image $x$, the $c_{\text{ref\_pose}}$ can be written as

$$c_{\text{ref}} = g_{\text{rp}}\left(f_{\text{s}}\left(x\right)\right),\tag{9}$$

where $f_{\text{s}}$ represents the skeleton extraction network and $g_{\text{rp}}$ denotes the reference pose guider.

### 4.4 SKELETAL DILATION MAP

Skeletal dilation map is used to mask the character and separate it from the background, and Fig. 4 illustrates its calculation. The skeletal dilation map does not perfectly cover the character area compared to segmentation maps, but it achieves better results. Using segmentation map as mask strictly requires highly accurate body segmentation maps as driving signals. However, given only an inference image and pose sequence during inference, it is challenging to generate a segmentation map sequence that perfectly aligns with the characters in the reference image. The discrepancies in segmentation map accuracy between training and inference lead to model misalignment and poor performance. In contrast, skeletal dilation maps can be generated directly from pose sequences and generalize well to characters of varying heights and body types. For both training and inference, we use the same skeletal dilation maps, enabling the model to learn to adapt its mask range and become less sensitive to mask precision. Further details can be found in the appendix (Sec. B.3).

### 4.5 MODEL ARCHITECTURE

Fig. 3 shows our framework. Specifically, the reference image is compressed using VAE and then fed into ReferenceNet (Cao et al., 2023). It interacts with the features in the U-net model through spatial cross-attention. Meanwhile, the CLIP-encoded reference image is used as the prompt to replace the text prompt in the diffusion model. The three proposed conditions and the pose drive signal sequence are encoded by the same structured guider, which consists of convolutional layers. These encoded features are then incorporated into the initial noise latent. The whole training objective can be formulated as

$$c_{\text{multi}} = c_{\text{pose}} + c_{\text{flow}} + c_{\text{depth}} + c_{\text{ref}},\tag{10}$$

$$\mathcal{L} = \mathbb{E}_{\mathcal{E}(v_{1:n}),c,\epsilon_{1:n}\sim\mathcal{N}(0,I),t}\left[||\epsilon - \epsilon_\theta(z_{1:n}^t,t,x,c_{\text{multi}})||_2^2\right],\tag{11}$$

where $c_{\text{multi}}$ denotes the overall control condition. The definitions of $v_{1:n}$, $\epsilon_{1:n}$, and $z_{1:n}^t$ are analogous to Eq. 1.

Table 3: Quantitative comparison on *Multi-Character* Bench. The best and the second-best results are indicated in red and blue respectively. AnimateAnyone† is trained on our noisy dataset.

| Method | FID↓ | SSIM↑ | PSNR↑ | LPIPS↓ | L1↓ | FID-VID↓ | FVD↓ |
|---|---|---|---|---|---|---|---|
| DisCo (Wang et al., 2023) | 77.61 | 0.793 | 29.65 | 0.239 | 7.64E-05 | 104.57 | 1367.47 |
| DisCo+ (Wang et al., 2023) | 73.21 | 0.799 | 29.66 | 0.234 | 7.33E-05 | 92.26 | 1303.08 |
| MagicAnime (Xu et al., 2023) | 40.02 | 0.819 | 29.01 | 0.183 | 6.28E-05 | 19.42 | 223.82 |
| MagicPose (Chang et al., 2023) | 31.06 | 0.806 | 31.81 | 0.217 | 4.41E-05 | 30.95 | 312.65 |
| Moore-AnimateAnyone | 33.04 | 0.795 | 31.44 | 0.213 | 5.02E-05 | 22.98 | 272.98 |
| AnimateAnyone† | 35.59 | 0.796 | 31.10 | 0.208 | 4.87E-05 | 22.74 | 236.48 |
| Ours | 26.95 | 0.830 | 31.86 | 0.173 | 4.01E-05 | 14.56 | 142.76 |

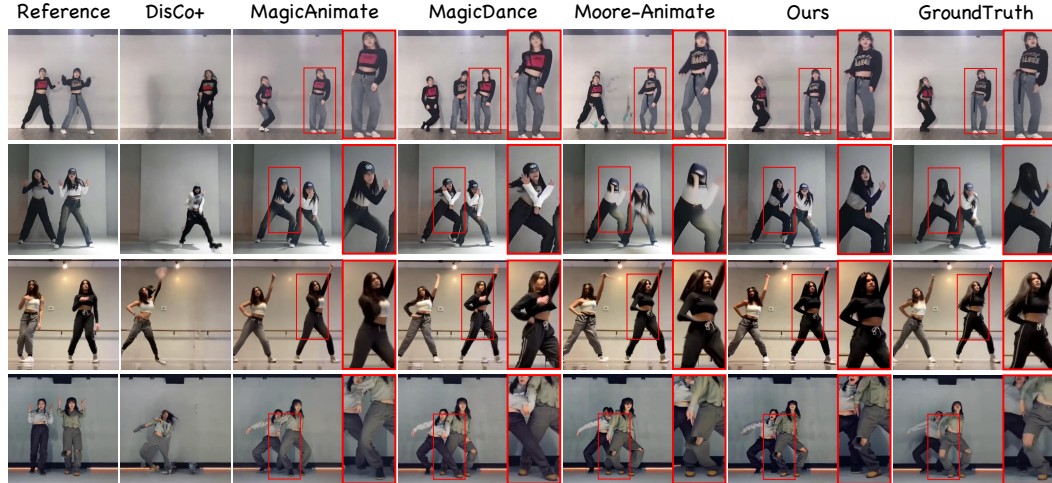

Figure 7: Qualitative comparisons on the *Multi-Character* bench.

# 5 EXPERIMENT

**Dataset.** The robustness of our model to noisy data enables direct training on unfiltered videos. We collect 4000 character-action videos of 2M frames as our training set. We analyze the level of contamination in the noisy dataset in the appendix (Sec. A.1).

**Training strategy.** Our model is trained in two stages. In the first stage, we freeze the VAE (Van Den Oord et al., 2017) and CLIP (Radford et al., 2021) image encoder, and remove the temporal motion modules. U-Net, ReferenceNet, and multiple-condition guiders are trained to align their spatial generative capacities. In addition, we utilize the weights of Stable Diffusion v1.5 (Rombach et al., 2022) to initialize this training stage. In the second stage, we incorporate temporal motion modules along with all parameters from stage one, aiming to endow the model with temporal smoothness. Besides, we use the weights of AnimateDiff v2 (Guo et al., 2023b) to initialize this training stage.

**Implementation Details.** We sample 16 frames of video, resize and center-crop them to a resolution of $896 \times 640$. Experiments are conducted on 8 NVIDIA A800 GPUs. Both stages are optimized using Adam with a learning rate $1 \times 10^{-5}$. In the first stage, we train our model for 60K steps with a batch size of 4, and in the second stage, we train for 60K steps with a batch size of 1. At inference, we apply DDIM (Song et al., 2020) sampler for 50 denoising steps, with classifier-free guidance (Ho & Salimans, 2022) scale of 1.5. See the appendix (Sec. C.1) for more details.

## 5.1 COMPARISONS

**Dataset and metrics.** Following the previous methods (Wang et al., 2023), we evaluate our method on TikTok videos (Wang et al., 2023) and TED-talks (Siarohin et al., 2021). Additionally, due to the lack of benchmarks for multiple-character video generation, we collect 20 multiple-character dancing videos with 3917 frames, named *Multi-Character*. This dataset serves as a benchmark for evaluating models' capabilities in generating pose-controllable videos with multiple characters. Our evaluation metrics adhere to the existing research literature. Specifically, we employ conventional image metrics to assess the quality of individual frames, including L1 error, PSNR (Hore & Ziou, 2010), SSIM (Wang et al., 2004), LPIPS (Zhang et al., 2018), and FID (Heusel et al., 2017). For

Table 4: Quantitative ablation results on TikTok Dataset.

| Method | FID↓ | SSIM↑ | PSNR↑ | LPIPS↓ | L1↓ | FID-VID↓ | FVD↓ |
|---|---|---|---|---|---|---|---|
| w/o. All Conditions | 54.42 | 0.685 | 29.01 | 0.316 | 1.06E-04 | 47.93 | 236.28 |
| w/o. Optical Flow | 52.94 | 0.710 | 28.21 | 0.318 | 0.96E-04 | 34.35 | 178.48 |
| w/o. Ref. Pose | 34.74 | 0.740 | 29.13 | 0.285 | 0.82E-04 | 19.58 | 139.32 |
| w/o. Depth Order | 27.43 | 0.754 | 29.98 | 0.270 | 0.79E-04 | 14.50 | 119.26 |
| Ours | 27.70 | 0.760 | 29.70 | 0.272 | 0.73E-04 | 14.30 | 117.81 |

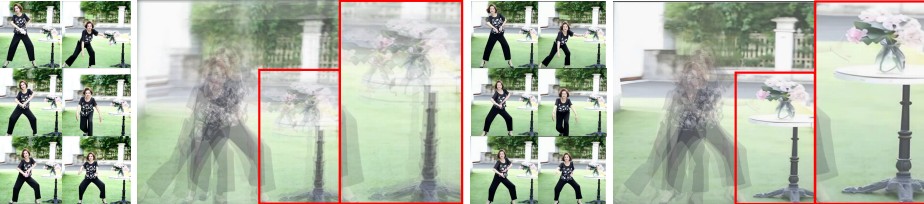

w/o. Optical Flow Condition        w/. Optical Flow Condition

Figure 8: Qualitative comparison results of ablation variants without optical flow condition. Transparent and overlay the images to clearly see the changes in the background.

video evaluation metrics FID-VID (Balaji et al., 2019) and FVD (Unterthiner et al., 2018), we form a sample by concatenating each consecutive 16 frames.

**Counterparts.** We compare with several state-of-the-art methods for character image animation. (1) MRAA (Siarohin et al., 2021) and TPSMM (Zhao & Zhang, 2022) are GAN-based methods. (2) DreamPose (Karras et al., 2023), Disco (Wang et al., 2023), MagicAnimate (Xu et al., 2023), MagicPose (Chang et al., 2023) and AnimateAnyone (Hu et al., 2023) are VLDM-based methods. Note that we evaluate the AnimateAnyone reproduced by MooreThreads (MooreThreads, 2024) on TED-talks and *Multi-Character*. (3) We reproduce AnimateAnyone trained on the noisy dataset.

**Unified evaluation standards.** We notice that not all approaches adhere to a uniform generation size. As methods with different generation sizes yield different metric results, potentially leading to unfair comparisons, we standardize the generation sizes by center-cropping and resizing to $512 \times 512$. Under this unified standard, we reevaluate methods that do not conform to this generation size and directly refer to the original results of methods that comply.

**Evaluation on TikTok dataset.** Table 1 presents the quantitative comparison of the TikTok dataset. Our model performs best on five metrics and second best on the FID and LPIPS metrics. Notably, our model excels particularly in video metrics FID-VID and FVD. Compared with the second-best method in FID-VID and FVD, our model shows a significant improvement of 39% and 35% respectively. The AnimateAnyone† we reproduce is trained on the noisy dataset and performs poorly, showing that noisy data greatly weakens the model. In contrast, the module we design enables our model to exhibit strong robustness to noisy data and perform excellently when trained on noisy data. Additionally, Ours* trained on Tiktok training set shows slight improvement over baselines, indicating our method excels when trained on clean data. The visual comparison is shown in the top three rows of Fig. 6. Our approach performs better in pose following and visual quality. Specifically, the first and third rows show that our method not only accurately follows the pose but also generates hand details. The second row suggests the powerful pose-following capability of our method, which is the sole approach capable of accurately generating the pose with the arm raised in reverse.

**Evaluation on TED-talks dataset.** As reported in Table 2, our model achieves SOTA performance across all metrics except PSNR. Compared with MagicPose achieving the second-best of FID-VID and FVD, our model demonstrates a substantial enhancement of 49% and 48% respectively. Again, it is observed that the base model performs poorly for the influence of noisy data. The bottom two rows in Fig. 6 illustrate the visual comparison. In the fourth row, only our method successfully maintains the consistency of the character holding white notes in the inference image. In the last row, our method exhibits the best performance in the pose following, with the fewest artifacts.

**Evaluation on *Multi-Character* bench.** Our results in Table 3 significantly outperform others across all metrics on *Multi-Character* bench. In terms of the video metrics FID-VID and FVD, we outperform the second-best method by 25% and 36%, respectively. The qualitative comparisons in Fig. 7 highlight the significant advantages of our approach in maintaining identity consistency (top row), pose following (second & third rows), and preserving spatial relationships (bottom row).

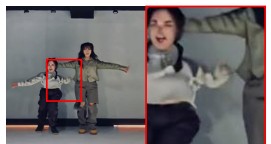 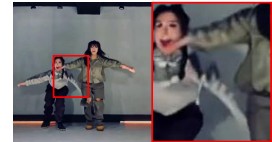 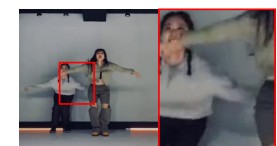

w/o. Depth Order Condition          w/. Depth Order Condition          GroundTruth

Figure 9: Qualitative comparison results of ablation variants without depth order condition.

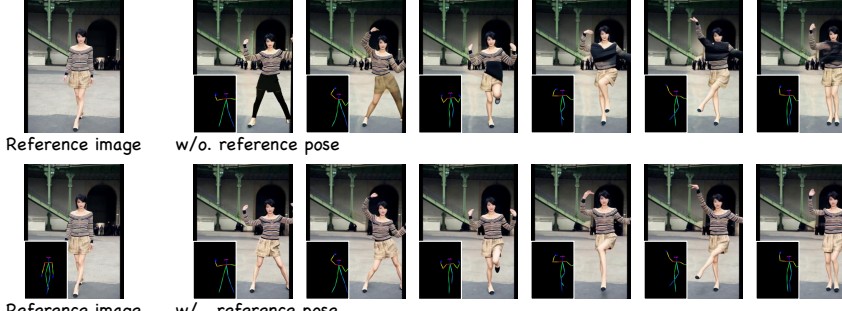

Reference image    w/o. reference pose

Reference image    w/. reference pose

Figure 10: Qualitative comparison results of ablation variants without reference pose condition.

## 5.2 ABLATION STUDY

To investigate the roles of the proposed conditions, we examine three variants without *Optical Flow*, *Depth Order*, and *Reference Pose*, respectively. In Table 4, the proposed full method ("Ours") with three conditions outperforms other variants, demonstrating the effectiveness of the three conditions.

**Optical flow.** In Table 4, the variant without optical flow exhibits the most significant decline, suggesting that optical flow has the most pronounced positive impact on the model. The comparison results shown in Fig. 8 illustrate where its primary effects are background stability. We overlay the transparent six frames to form the right image, enabling a clear depiction of the moving region. As indicated by the red boxes, the background in the results of the variant without optical flow shows noticeable shaking, whereas the background in the results of the full method remains consistently stable. This validates that noisy data with unstable backgrounds leads to the generation of unstable backgrounds, whereas the incorporation of the optical flow condition can solve this problem.

**Depth order.** Fig. 9 validates the positive impact of the depth order condition. The red boxes highlight the overlapping areas of multiple characters. The variant without depth order condition fails to generate the hand of the character on the left placed behind the right character, instead generating a malformed hand. Conversely, the full method with the depth order condition generates overlapping areas of multiple characters, *i.e.*, the hand of the left character is positioned behind the right one, and the hand of the right character is in front of the head of the left one.

**Reference pose.** Fig. 10 illustrates the comparison when the character position of the inference image and pose sequence are not aligned. The full method with reference pose performs better in terms of image quality and character consistency. The variant without reference pose maintains the background consistency, while failing to maintain character consistency (see the changes of clothing of the character). In contrast, the full method with reference pose effectively achieves alignment between character and pose sequence, thereby maintaining character and background consistency.

## 6 CONCLUSION

In this paper, we design three guiders to enhance the implicit decoupling ability of a pose-controllable character image animation framework that integrates multiple conditions, addressing the unstable background and poor handling of body occlusions in multiple character scenes. The optical flow guider decouples the background to facilitate the learning of stable background generation. The depth order guider decouples multiple character features into individuals to solve the problem of multiple character generation. The reference pose guider enhances the learning of characters' appearance. Moreover, we have curated and released a benchmark dataset of pose-controllable videos with multiple characters. Experiment studies show the effectiveness of our method.

ACKNOWLEDGMENT

This work was supported in part by the National Natural Science Foundation of China (Grant No. 62372480), in part by the Guangdong Basic and Applied Basic Research Foundation (No. 2023A1515012839), in part by Huawei Gift Fund (No. HUAWEI25IS02), and in part by HKUST-MetaX Joint Lab Fund (No. METAX24EG01-D).

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

## APPENDIX

In this supplementary material, we present:

## A  DATASET SUPPLEMENT

### A.1  ANALYSIS OF NOISY TRAINING DATASET

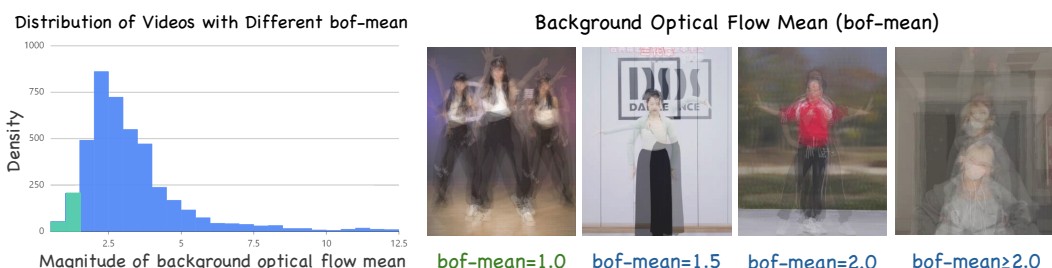

Figure 11: Distribution histogram and image examples of background optical flow mean.

We analyze the level of contamination in the noise dataset. Specifically, we calculate the background optical flow using a skeletal dilation mask to exclude character regions, then average it across frames to derive the background optical flow mean for each video. Fig. 11 (right) shows videos with different background optical flow means. It is observed that only when the background optical flow mean is at least less than 1, the motion of the background of the video is imperceptible to human eyes. Fig. 11 (left) illustrates the distribution of the background optical flow mean in the training set. It indicates that only about 12% of the training set videos have a background optical flow mean of less than 1. A noise data proportion of 88% indicates severe contamination of the dataset. Therefore, it is necessary to incorporate background optical flow maps into the network for training.

We randomly sample 400 videos (approximately 10% of the dataset) from our dataset and analyze the distribution of videos based on the count of characters they contain. Fig. 12 (left) illustrates the proportion of videos regarding different numbers of characters in the sampling set. Specifically,

Figure 12: Sampling distribution of character counts in videos and body occlusion rates in double-character videos.

single-character videos account for $66.25\%$, double-character videos account for $29.5\%$, and videos of triple characters and above account for a total of $4.25\%$. It can be concluded that the majority of videos in our dataset are single-character or dual-character, with very few videos containing three or more characters.

Subsequently, we compute the body occlusion rates among characters across all sampled multi-character videos. Specifically, we calculate the intersection area of the skeletal dilation maps for multiple characters in each frame of the video, and then divide it by the union area of these skeletal dilation maps for the same frame. This yields the body occlusion rate for that frame. Subsequently, we compute the average body occlusion rate across all frames of a video and analyze the distribution across all videos. Fig. 12 (right) illustrates the distribution of body occlusion rates in multi-character videos. We can find that $25\%$ of the videos have a body occlusion rate within $0.05$, $25\%$ fall between $0.05$ and $0.13$, $25\%$ are between $0.13$ and $0.21$, and the remaining $25\%$ have body occlusion rates exceeding $0.21$. It indicates that, on average, the body occlusion rates of most multiple-character videos fall between $0$ and $0.21$, with videos exhibiting high occlusion rates above $0.21$ being relatively rare. Meanwhile, we directly observe that there are many multi-character videos with body occlusion rates around $0$.

## A.2 DETAILS OF THE TRAINING DATASET

Table 5: The detailed composition of the training dataset.

| Source | Videos | Frames | Proportion |
|---|---|---|---|
| Tiktok | 2,493 | 1,379,449 | 68.5% |
| YouTube | 938 | 435,293 | 21.6% |
| Kuaishou | 424 | 115,101 | 5.7% |
| Bilibili | 162 | 83,785 | 4.2% |

We collect $4,017$ character videos with the amount of $2,013,628$ frames as our training set. The data come from public videos on TikTok, YouTube, and other websites. The detailed composition of the training dataset is shown in Table 5.

## A.3 DETAILS OF THE MULTI-CHARACTER BENCHMARK

We collect 20 multiple-character dancing videos of 3917 frames in total, from social media, named *Multi-Character*. Table 6 shows the detailed sources of *Multi-Character*.

Table 6: The source of Multi-character benchmark.

| Video Name | Url | Timestamp |
|---|---|---|
| Daovm348PQQ_0 | https://www.youtube.com/watch?v=Daovm348PQQ | 00:10–00:14 |
| Daovm348PQQ_1 | https://www.youtube.com/watch?v=Daovm348PQQ | 00:16–00:25 |
| Daovm348PQQ_2 | https://www.youtube.com/watch?v=Daovm348PQQ | 00:47–00:53 |
| Daovm348PQQ_3 | https://www.youtube.com/watch?v=Daovm348PQQ | 01:11–01:15 |
| Daovm348PQQ_4 | https://www.youtube.com/watch?v=Daovm348PQQ | 01:16–01:21 |
| Daovm348PQQ_5 | https://www.youtube.com/watch?v=Daovm348PQQ | 01:22–01:25 |
| Daovm348PQQ_6 | https://www.youtube.com/watch?v=Daovm348PQQ | 02:02–02:09 |
| Daovm348PQQ_7 | https://www.youtube.com/watch?v=Daovm348PQQ | 02:11–02:15 |
| Daovm348PQQ_8 | https://www.youtube.com/watch?v=Daovm348PQQ | 02:16–02:20 |
| HpFDXGAo25c_0 | https://www.youtube.com/watch?v=HpFDXGAo25c | 00:20–00:33 |
| HpFDXGAo25c_1 | https://www.youtube.com/watch?v=HpFDXGAo25c | 00:38–00:43 |
| HpFDXGAo25c_2 | https://www.youtube.com/watch?v=HpFDXGAo25c | 00:44–00:52 |
| jx_VseYOi5A_0 | https://www.youtube.com/watch?v=jx_VseYOi5A | 00:32–00:37 |
| jx_VseYOi5A_1 | https://www.youtube.com/watch?v=jx_VseYOi5A | 00:40–00:53 |
| jx_VseYOi5A_2 | https://www.youtube.com/watch?v=jx_VseYOi5A | 01:02–01:07 |
| jx_VseYOi5A_3 | https://www.youtube.com/watch?v=jx_VseYOi5A | 01:08–01:12 |
| ka3BfUsvRqE_0 | https://www.youtube.com/watch?v=ka3BfUsvRqE | 00:21–00:27 |
| ka3BfUsvRqE_1 | https://www.youtube.com/watch?v=ka3BfUsvRqE | 00:28–00:33 |
| ka3BfUsvRqE_2 | https://www.youtube.com/watch?v=ka3BfUsvRqE | 02:56–03:05 |
| ycInNCB8rbA_0 | https://www.youtube.com/watch?v=ycInNCB8rbA | 00:10–00:15 |

## B  METHODOLOGY SUPPLEMENT

### B.1  PSEUDOCODE OF DEPTH ORDER GUIDER

---

**Algorithm 1:** Pseudocode for depth order map mask extraction

---

**Input:** Given a training video $v$ of length $N$, where the $i$-th frame is denoted as $v_i$, and suppose that there are $J$ characters on $v_i$. The skeleton extraction network is denoted as $f_\mathrm{s}$. The expansion network is denoted as $f_\mathrm{e}$. The depth extraction network is denoted as $f_\mathrm{d}$. The average depth sorting (ascending order) operator is denoted as $f_\mathrm{sort}$. The value assigned to $r_j$ based on depth ranking is denoted as $L_{r_j}$. The depth guider is denoted as $g_\mathrm{dp}$.

1  Initialize $c_{\mathrm{depth},1,0}, \ldots, c_{\mathrm{depth},N,0} = \mathbf{0}$.
2  Initialize an array $C_R$.
3  **for** $i = 1$ **to** $N$ **do**
4  $\quad$ $a_{i,1}, \ldots, a_{i,J} = f_\mathrm{e}\left(f_\mathrm{s}\left(v_i\right)\right)$
5  $\quad$ **for** $j = 1$ **to** $J$ **do**
6  $\qquad$ $m_{i,j} = a_{i,j} - \left(1 - \bigcup_{j \in \{1,\ldots,J\}}\left(a_{i,j}\right)\right)$
7  $\quad$ $r_1, \ldots, r_J = f_\mathrm{sort}\left(m_{i,1} \odot f_\mathrm{d}\left(v_i\right), \ldots, m_{i,J} \odot f_\mathrm{d}\left(v_i\right)\right)$
8  $\quad$ **for** $r_j = r_1$ **to** $r_J$ **do**
9  $\qquad$ $c_{\mathrm{depth},i,r_j} = m_{i,r_j} \odot L_{r_j} + \left(\left(1 - m_{i,r_j}\right) \odot c_{\mathrm{depth},i,(r_j - 1)}\right)$
10  $\quad$ $C_R\left[i\right] \leftarrow c_{\mathrm{depth},i,r_J}$
11  $c_\mathrm{depth} = g_\mathrm{dp}\left(C_R\right)$
12  **return** $c_\mathrm{depth}$

---

### B.2  LONG VIDEO INFERENCE

We employ the overlap method for long video inference to maintain the consistency of long video. As illustrated in Fig. 13, we divide the pose sequence into multiple shorter segments for inference, with overlapping parts between adjacent segments. For the overlapping parts of two segments, we

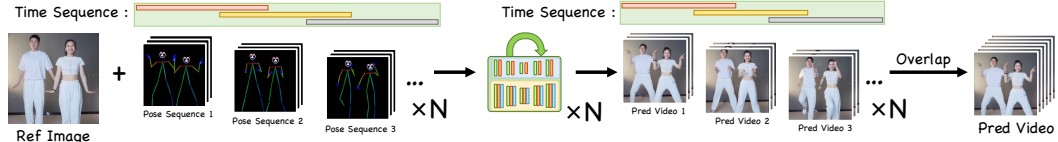

Figure 13: The pipeline for inferring long videos.

perform addition and averaging to generate temporal smoothing between the two segments. In this work, we perform inference on every 16 frames with a stride of 8 frames, and then stitch them together using an overlap of 8 frames.

## B.3   FURTHER DETAILS OF SKELETAL DILATION MAP

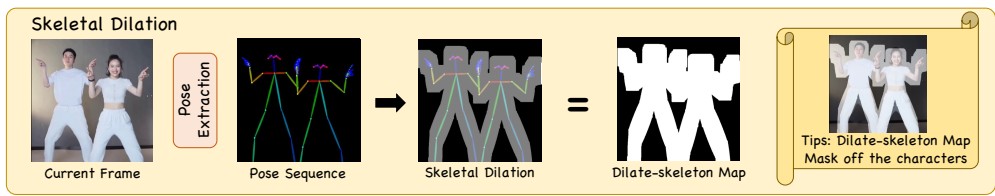

Figure 14: The pipeline for calculating skeletal dilation map.

We use skeletal dilation map as mask to cover the character region in the "optical flow maps" and "depth order maps". The purpose of masking off the character regions in "Optical Flow Maps" is to prevent character motion from affecting the decoupling of the background. Additionally, the skeletal dilation map is used to refer to different character areas in "Depth Order Maps", facilitating the decoupling of characters into individual ones.

Fig. 14 shows the pipeline for calculating the skeletal dilation map, which is directly generated from the pose frame. Maybe one concern is that skeletal dilation map often does not cover the character region comprehensively. However, in our attempts, the skeleton dilation map achieves better results compared to the segmentation map that fully covers the human body. We analyze the reasons as follows: (1) The consistency between training and inference processes enables the model to learn to ignore the deviations of skeleton dilation maps. Specifically, skeletal dilation map represents only rough but not precise character regions, during both training and inference. When using the skeletal dilation map as mask during training, the model learns to map rough character regions to precise character regions and applies this capability during inference. Similarly, research on the "regional image animation" task and "modify region animation" task (Ma et al., 2024b) have confirmed that models animate the objects represented by the mask region, rather than animating the mask region itself. Fig. 15 (a) shows that when the skeleton dilation map does not perfectly cover the character, the character still separates perfectly from the background, and the character animation remains outstanding. Fig. 15 (b) illustrates that the parts of the character that exceed the coverage range of the skeletal dilation map can also be animated perfectly. In contrast, Fig. 15 (c) shows a bad case where the portion of the clothing extending beyond the coverage of the skeletal expansion map is too large, resulting in poor animation effects for the clothing. (2) The use of the skeleton dilation map exhibits high tolerance and insensitivity to small amounts of noise. Specifically, the skeleton dilation map requires the model to learn to adapt its mask range, resulting in less sensitivity to mask precision. This enables the same skeletal dilation map to perform well across various scenarios involving different body types and clothing, demonstrating strong generalization capability. In contrast, the use of segmented images imposes strict requirements for high accuracy, and segmentation maps with different body types or clothing, or even slightly different fingers can lead to significant degradation. (3) The discrepancies in segmentation map accuracy between training and inference lead to model misalignment and poor performance. During training, using the segmentation map generated from the original video will make the model highly sensitive to errors. However, given only the

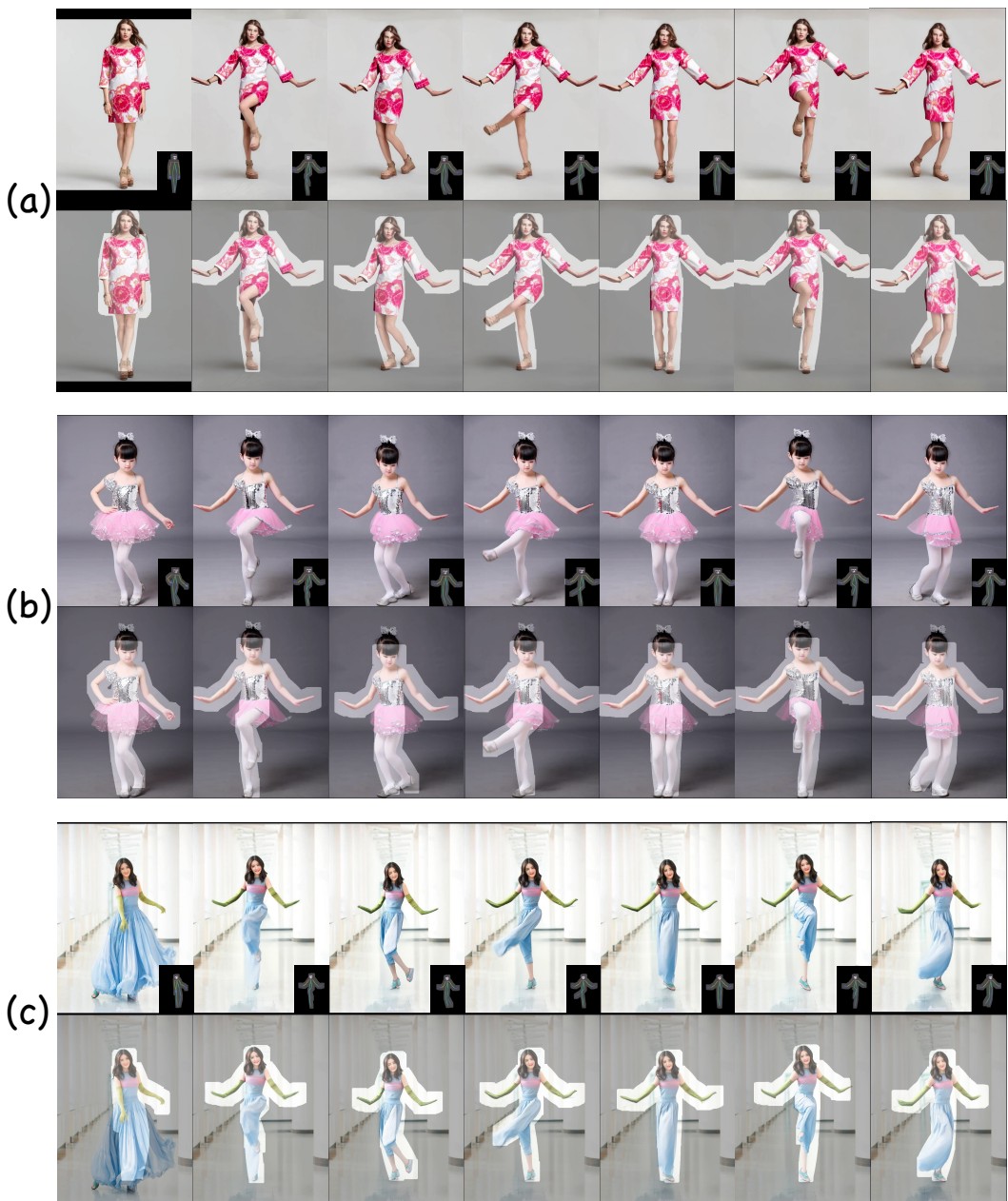

Figure 15: The results using skeletal dilation map as mask.

inference image and pose sequence during inference, it is challenging to generate a segmentation map sequence that perfectly aligns with the characters in the reference image. Therefore, minor misalignments in the inference segmentation map can lead to significant degradation of the results.

## C    EXPERIMENTAL SUPPLEMENT

### C.1    MORE IMPLEMENTATIONS

In the data processing, we utilize the DWPose (Yang et al., 2023) to extract pose sequence from videos, and PWC-Net (Sun et al., 2018) from the open-source toolbox MMFlow (OpenMMLab,

Table 7: Inconsistent inference standards across all methods.

| Method | Inference Size | Center-Crop | Uninterrupted Frames |
|---|---|---|---|
| MRAA (Siarohin et al., 2021) | 384×384 | ✗ | ✓ |
| TPSMM (Zhao & Zhang, 2022) | 384×384 | ✗ | ✓ |
| DreamPose (Karras et al., 2023) | 512×640 | ✗ | ✓ |
| DisCo (Wang et al., 2023) | 256×256 | ✓ | ✗ |
| MagicAnimate (Xu et al., 2023) | 512×512 | ✓ | ✓ |
| MagicPose (Chang et al., 2023) | 512×512 | ✗ | ✓ |

Table 8: The ablation experiment on the inference standards of Disco+.

| Inconsistent Standards | FID↓ | SSIM↑ | PSNR↑ | LPIPS↓ | FID-VID↓ | FVD↓ |
|---|---|---|---|---|---|---|
| origin size, w/o center-crop | 30.75 | 0.668 | 29.03 | 0.292 | 59.90 | 292.80 |
| 256×256, w/o uninterrupted frames | 28.31 | 0.674 | 29.15 | 0.285 | 55.17 | 267.75 |
| 512×512, w/o uninterrupted frames | 48.29 | 0.713 | 28.78 | 0.320 | 52.56 | 334.67 |
| 512×512, w/ uninterrupted frames | 48.29 | 0.713 | 28.78 | 0.320 | 47.73 | 312.49 |

2021) to calculate optical flow vectors. Additionally, we use the Depth Anything (Yang et al., 2024) to extract depth maps from videos.

When conducting long video inference, we perform inference on every $16$ frames with a stride of $8$ frames, and then stitch them together using an overlap of $8$ frames. Besides, we resize and center-crop the "Reference Image" and "Pose Sequence" to a uniform resolution of $896 \times 640$ pixels ($512 \times 512$ pixels in comparative experiments). We apply the DDIM sampler for $50$ denoising steps, with classifier-free guidance.

Regarding the inference resources, our method requires 24GB of VRAM to generate a $640 \times 896$ video, 16GB of VRAM for a 480P ($480 \times 854$) video, and 12GB of VRAM for a 360P ($360 \times 640$) video. Thus, our model can run on most commodity-level GPUs with the necessary adjustment to an appropriate video generation resolution.

## C.2 UNIFIED STANDARD FOR COMPARATIVE EXPERIMENTS

We notice in the comparative experiment that not all approaches adhere to a uniform inference size and other inference details. As depicted in Table 7, there are primarily three inconsistent standards that may affect fair comparisons. In the following, the method Disco+ will be used as an example to illustrate how inconsistent standards affect the fair comparison of metrics, as shown in Table 8. First, Table 7 shows some works resize without center-crop, which can result in significant differences in the test set, as illustrated in Table 8. Second, as shown in the comparison between the second and third rows of Table 8, different inference sizes result in different metric values. Table 7 shows the inconsistent inference sizes of each method. Third, compared to other methods, Disco has a bug in measurement, resulting in fewer video segments being sampled when calculating FID-VID and FVD. This will lead to a decrease in FID-VID and FVD of Disco, as shown in Table 8.

Because methods with different standards result in different metrics, potentially leading to unfair comparisons, we standardize the inference sizes by center-cropping and resizing to $512 \times 512$, and we fix the bug in the measurement of disco. Under this unified standard, we reevaluate methods that do not conform to this inference size and directly refer to the data from the original literature of methods that do comply. Most previous works directly refer to the inconsistent statistical data from other works for comparison, resulting in unfair comparisons. We are the first to conduct comparative work under a unified standard.

## D  LIMITATION

In this work, we are dedicated to addressing the challenges in pose-controllable multiple-character animation. However, there are still several problems we have not resolved. First, similar to most diffusion-based approaches, our model struggles to generate highly refined facial and hand details. Second, our model also struggles to generate substantial swaying of long skirts, Hanfu, or other large-area clothing very well, as shown in Fig. 15 (c). Third, our model also faces challenges in handling complex multiple-character scenarios, such as those involving four or more characters, or extensive swapping of positions among characters.

## E  MORE QUALITATIVE EXPERIMENTS

### E.1  ADDITIONAL COMPARISON

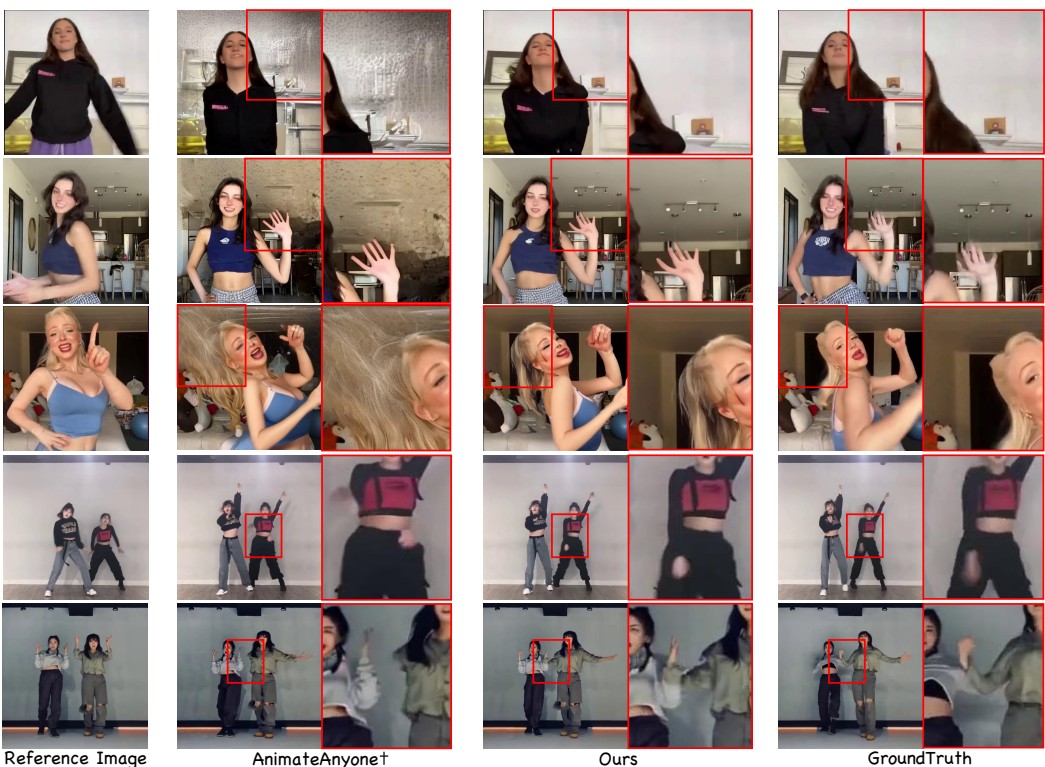

Figure 16: Qualitative comparison between ours and AnimateAnyone†.

We reproduce AnimateAnyone† on our noisy dataset, but achieve worse results compared to the original AnimateAnyone, as shown in Tables 1, 2, and 3. This indicates that our noisy dataset's poor video quality significantly weakens the model. Fig. 16 illustrates the qualitative comparison between ours and AnimateAnyone†. In the top three rows, it can be observed that the backgrounds generated by AnimateAnyone† exhibit numerous artifacts. In the bottom two rows, AnimateAnyone† loses arm when generating occluded body parts of multiple characters. In contrast, ours addresses these issues, demonstrating that the proposed conditions have a significant positive impact on the model.

Fig. 17 illustrates a comparison of facial regions between ours and baselines on the TikTok dataset. It can be observed that our method generates the optimal character face, particularly achieving consistency with the reference image in terms of expressions and hairstyles. Note that we use the reference image, rather than the groundtruth, as the correct sample.

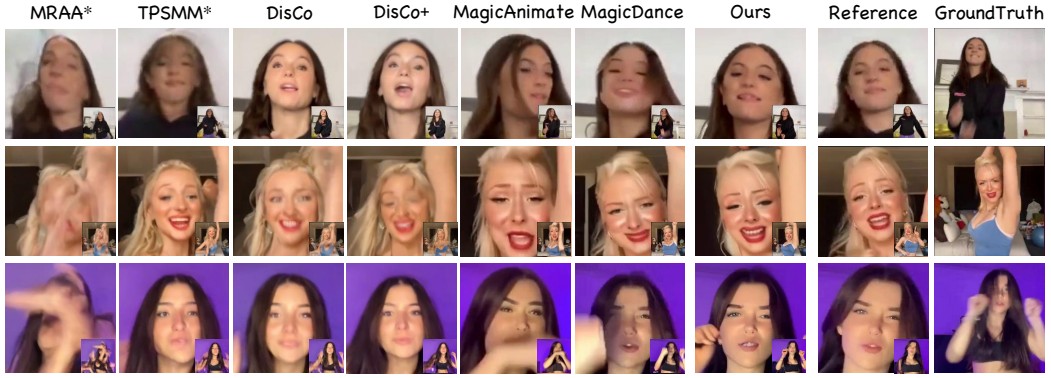

Figure 17: Qualitative comparison of facial regions.

## E.2 ADDITIONAL ABLATION

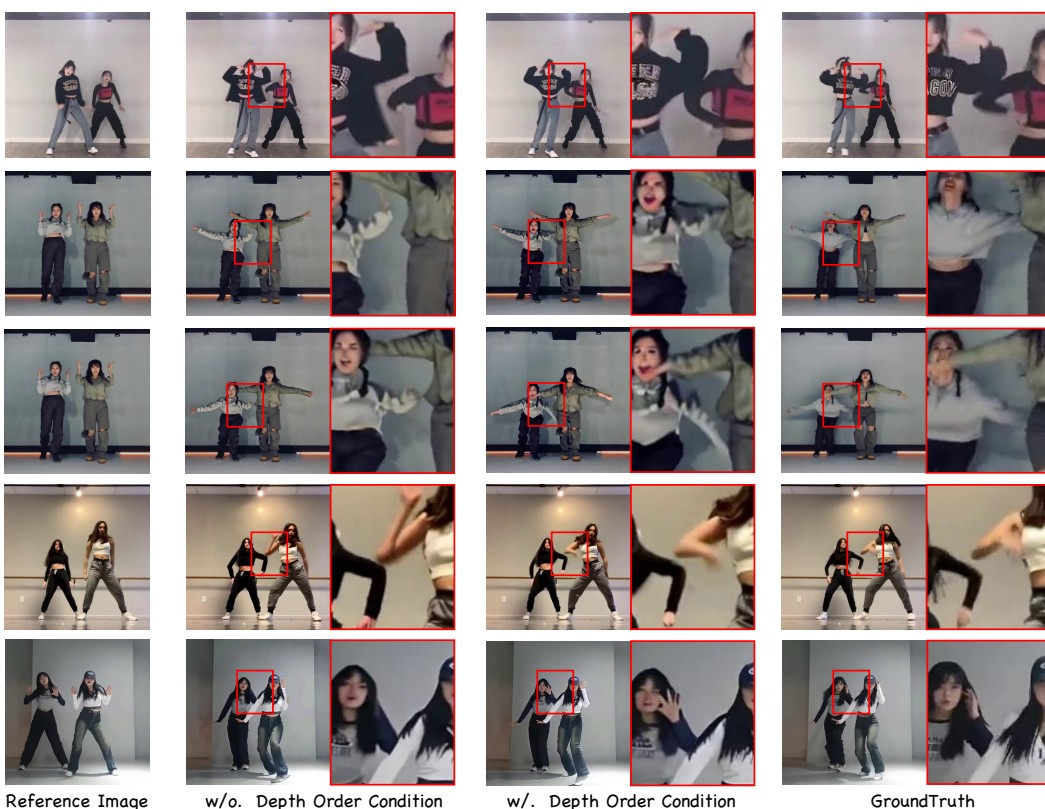

Figure 18: Additional visualizations of ablation variants without depth order condition.

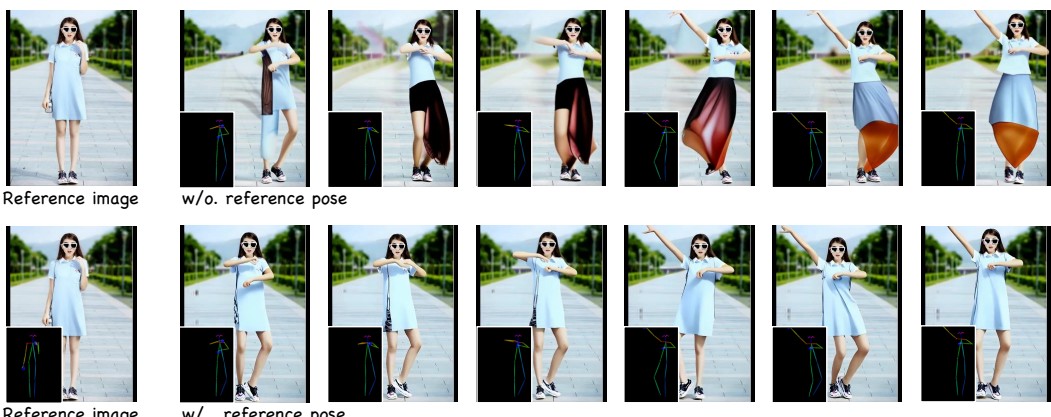

Figure 19: Additional visualizations of ablation variants without reference pose condition.

## E.3 ADDITIONAL VISUALIZATIONS

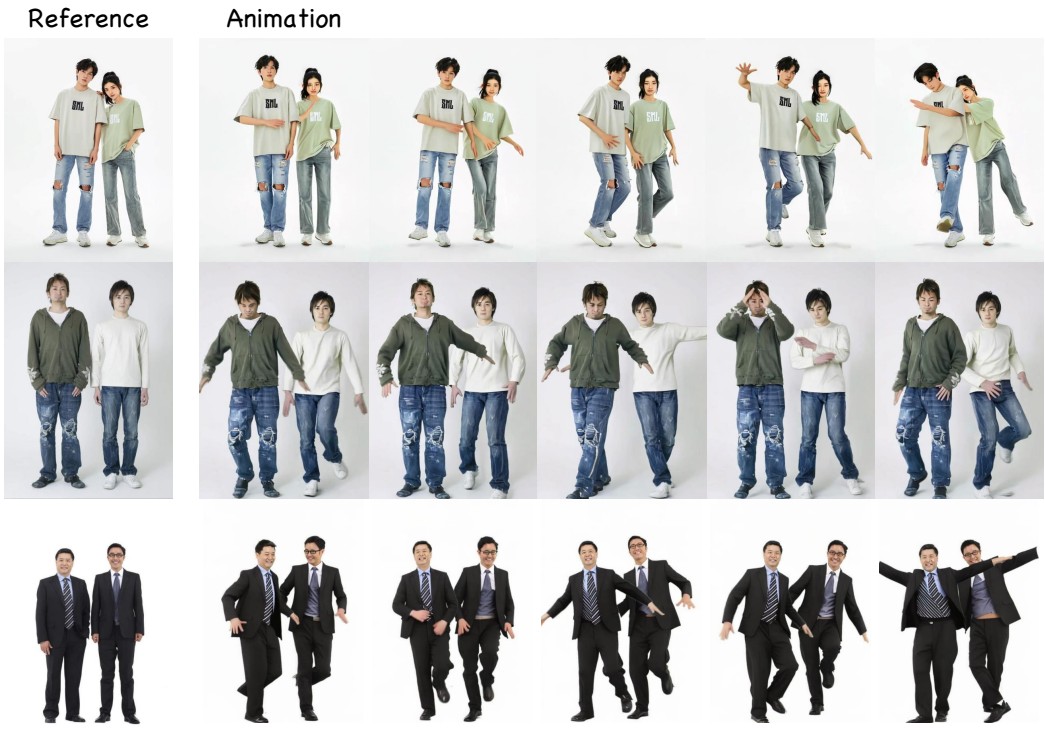

Figure 20: Additional visualizations, example 1, different poses for different characters.

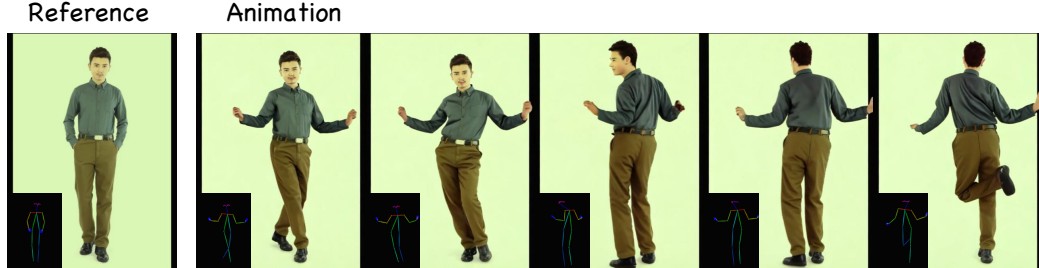

Figure 21: Additional visualizations, example 2, character rotation.

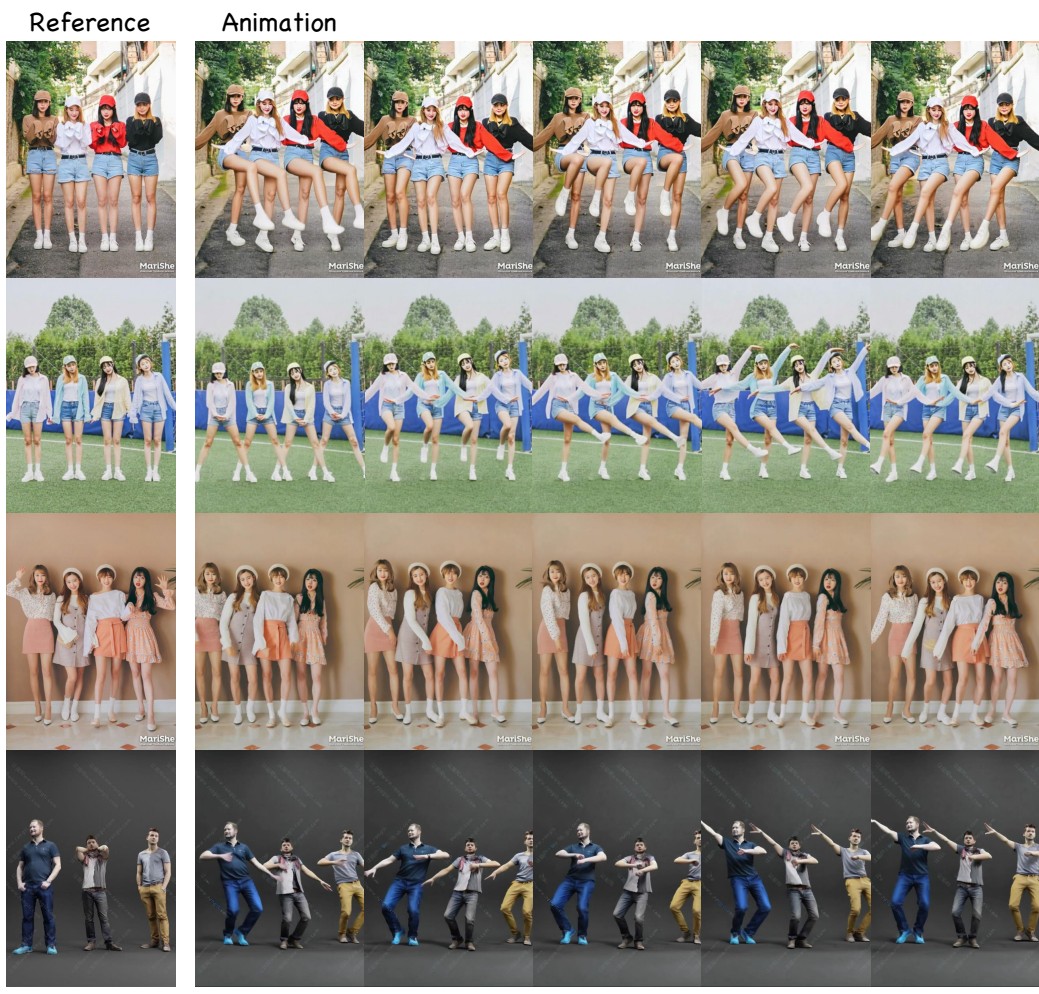

Figure 22: Additional visualizations, example 3, three or four characters animations.

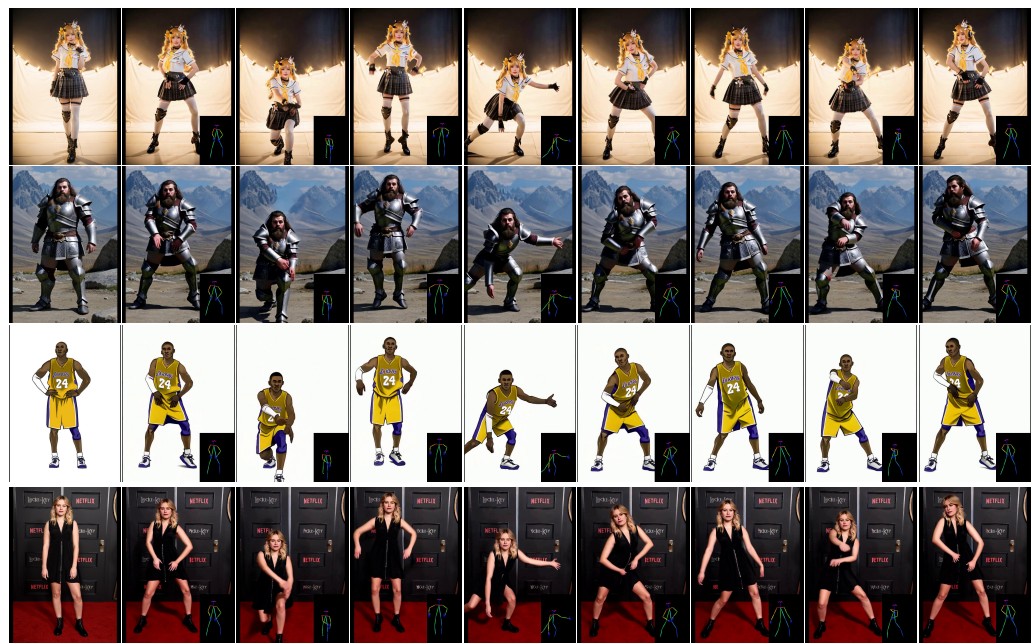

Figure 23: Additional visualizations, example 4, single character dance - Just Because You're So Beautiful.

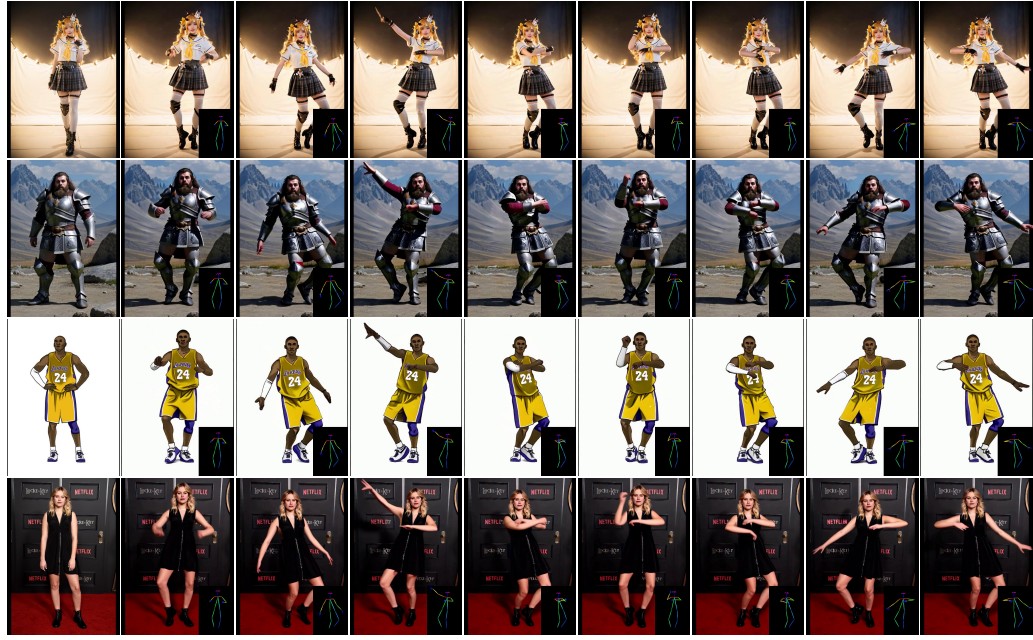

Figure 24: Additional visualizations, example 5, single character dance - the viral kemusan.

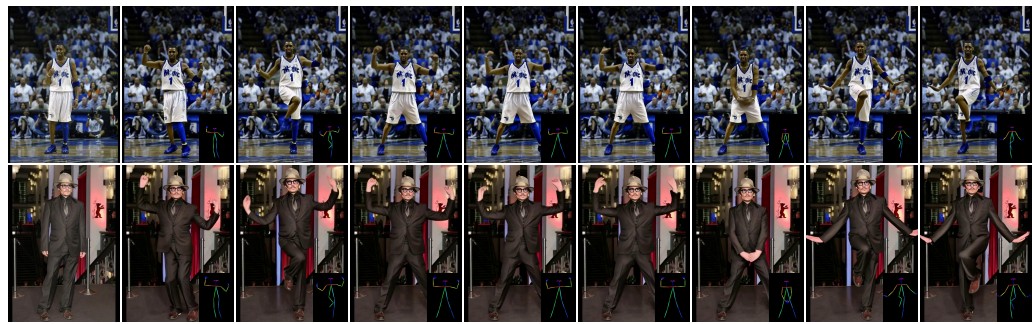

Figure 25: Additional visualizations, example 6, single character dance - the rabbit dance.

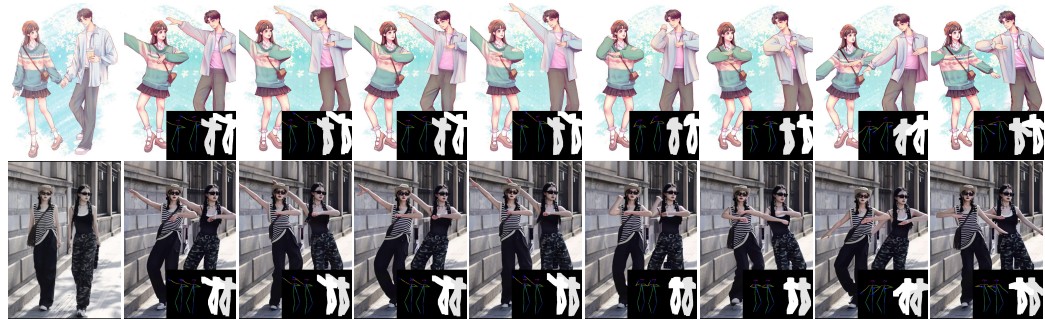

Figure 26: Additional visualizations, example 7, multiple character dance - the viral kemusan.

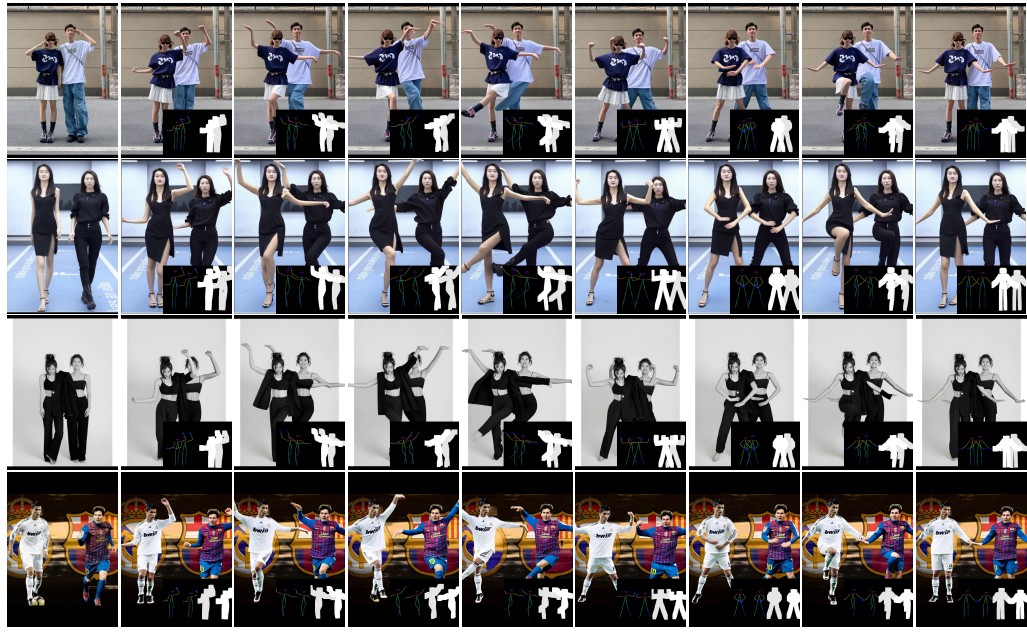

Figure 27: Additional visualizations, example 8, multiple character dance - the rabbit dance.

