# OpenReview forum: "Towards Multiple Character Image Animation Through Enhancing Implicit Decoupling"
_ICLR.cc/2025/Conference — ICLR 2025 Poster_

### Official Review · Reviewer_eUry · 2024-10-23

**Soundness:** 3
**Presentation:** 3
**Contribution:** 3
**Rating:** 6
**Confidence:** 3

**Summary:**

The paper tackles the problem of multiple-character image animation and addresses the failures in current approaches by introducing additional control signals: 1) To alleviate the influence of non-static background, it adds the optical flows as conditions; 2) To avoid identity mixup in multi-character animation, it adds depth order guidance that labels the identity id within the skeletal dilation map. It achieves comparable performance on single-character animation benchmarks and better quality on multi-character animation.

**Strengths:**

* The proposed methods are well-motivated. The authors demonstrate the effectiveness of the proposed control signals by comprehensive ablation studies.
* The skeletal dilation map is novel to me as an alternative to the segmentation mask. The authors explain the choice and its advantage over the segmentation mask, which is sound.
* It shows better quality on multi-character animation compared to baselines.

**Weaknesses:**

* The head poses are limited in all the qualitative results presented in the paper, including the figures and the supplementary videos. For example, in Fig. 1, the head poses of all characters are the same as the reference images while other parts of the body are retargeted into new poses. It makes it difficult to assess the animation quality without qualitative results showcasing more diverse head poses. This is my main concern since faces are probably the most difficult and important part of the animation. Comparing facial regions with the baselines will help the judgment.

* As mentioned in the limitations, the use of the skeletal dilation map restricts the animation of clothes that cover a significantly larger area than the skeletal dilation map. Is the kernel size of dilation a crucial factor for the method? How does it impact the animation of clothes?

**Questions:**

In addition to the weakness, can the authors clarify how the depth order map of the target pose sequences is acquired during inference? More specifically, how the depth order is determined given only the skeletons of multi-character poses?

---

> ### Author Response · Authors · 2024-11-22
> **Response to Reviewer eUry**
>
> **Q1**: Animation results of diverse head poses.
>
> **A1**: We have updated the supplementary video, including the results of obvious movement of head and diverse head poses (1:12–1:39 in the video). The skeleton in the pose map contains five skeletal points of the head, which clearly represent the posture and orientation of the head. Therefore, our method can generate various head movements and generate correct results in the scenery of various head poses in reference image.
>
> Furthermore, we have included a facial comparison with baselines in Appendix Section E.1 of the updated paper.
>
> &nbsp;
>
> **Q2**: Is the kernel size of dilation crucial for the method?
>
> **A2**: The kernel size is not the critical factor. As stated in Appendix B.3, the use of skeletal dilation maps ensures consistency during training and inference, which is the main reason why they outperform segmentation maps. If the kernel size is large, the model learns the alignment between characters and large skeletal dilation areas, otherwise, it learns the alignment with smaller skeletal dilation areas. The limitation in generating large-area clothing animations lies in their low proportion within the dataset. The model lacks sufficient samples to learn the correspondence between characters wearing various large-area clothing and the skeleton dilation areas. Consequently, increasing the kernel size does not help the generation of large-area clothing.
>
> &nbsp;
>
> **Q3**: How the depth order map of the target pose sequences is acquired during inference, and how the depth order is determined given only the skeletons of multi-character poses?
>
> **A3**: During inference, we use the reference image to determine the depth order. Specifically, we extract the depth map of the reference image and sort the characters based on their average depth values. Subsequently, we directly assign the order value from the reference image to the skeleton dilation areas of the corresponding characters. We determine the correspondence between the character and the skeleton dilation area by calculating the distances between the facial skeletal points of the reference image character and the facial skeletal points of the inference pose skeleton.
>
> Thank you for your question. In the revised version, we have provided a description of the depth information during inference (on page 6).

---

> > ### Comment · Reviewer_eUry · 2024-11-22
> >
> > Thanks for the clarification. The method is overall well-motivated and effective. Due to the flickering and distortions in the facial regions when the characters are animated (e.g., 0:11 in the new supplementary video), I would like to keep my original score.

---

> > > ### Author Response · Authors · 2024-11-23
> > > **Further Response to Reviewer eUry**
> > >
> > > Thanks a lot for your response and the acknowledgment that "the method is overall well-motivated and effective".
> > >
> > > For the concerned "flickering and distortions in the facial regions", we believe it is fair to discriminate between head pose and facial region.
> > >
> > > Diverse head poses mean various orientations or positions of a person's head referring to a centering head of the frontal face. In this scenario, we have already successfully animated these diverse head poses including the most challenging 360-degree rotation, as demonstrated by the supplement video.
> > >
> > > **Facial region animation** is another isolated task, e.g., talking face, due to its specific requirement of additional guidance, like reference facial video or facial landmark sequence. And in that kind of task, the size and the resolution of the face region are large. In our **full-body animation**  task, the face region is always small, and it is challenging to accurately reconstruct faces, particularly when the number of characters increases, and it is inherent to the variational autoencoder (VAE) framework we utilized. This is a known limitation of the latent stable diffusion model.
> > >
> > > We also notice that some commercial full-body animation applications, like Dance King for All which is based on Animate Anyone in Tongyi mobile app,  typically employ techniques such as video face-swapping. This strategy effectively enhances the facial representation without being constrained by the VAE's reconstruction capabilities. So it should not be taken as a drawback of full-body animation.
> > >
> > > We appreciate the opportunity to clarify these points, and sincerely hope you can re-evaluate our work considering these clarifications.

---

> > > > ### Comment · Reviewer_eUry · 2024-11-29
> > > >
> > > > Thank you for your efforts. I understand that facial regions are inherently challenging. I’ll keep my rating above the acceptance threshold.

---

> > > > > ### Author Response · Authors · 2024-11-30
> > > > >
> > > > > Dear Reviewer eUry,
> > > > >
> > > > > Thank you for your understanding and for recognizing the challenges associated with facial regions in our work. We sincerely appreciate your supportive feedback. Thank you once again for your time and valuable insights.
> > > > >
> > > > > Best regards,
> > > > >
> > > > > The Authors

---

### Official Review · Reviewer_TMXE · 2024-11-02

**Soundness:** 3
**Presentation:** 3
**Contribution:** 3
**Rating:** 6
**Confidence:** 4

**Summary:**

Existing human animation methods struggle with multi-character animation tasks and are difficult to train on data with unstable, noisy backgrounds. This paper addresses these challenges with two main motivations: to decouple background motion from static elements and to separate multiple characters. To this end, the authors propose a novel framework that includes an optical flow guider and a depth-order guider.

The optical flow guider encodes background motion, while the depth-order guider captures the spatial relationships among multiple characters. Additionally, a reference pose guider is introduced to encode the pose of a reference image, simplifying the textual remapping process.

Evaluations on multi character benchmark demonstrate that this method generates high-quality character animations when trained on noisy web-collected datasets.

**Strengths:**

- The paper identifies a key challenge in training human animation models on in-the-wild data—the instability of background motion—and proposes a pipeline to address this issue.

- An optical flow-based conditioning module is introduced, enabling human animation methods to train on data with background instability, which helps scale the approach to noisy, real-world datasets.

- The paper presents a heuristic depth-order map calculation pipeline, providing simple yet effective spatial information for multi-character generation tasks.

**Weaknesses:**

- Although the authors claim the method can handle multiple characters, most of the results feature only one or two characters.

- While the paper includes some analysis of background motion in the training dataset, it lacks in-depth analysis related to multiple characters, such as the effect of character count or occlusion ratios.

**Questions:**

- Can the authors provide animation results for more challenging scenarios, such as dense crowd animations or high occlusion ratios?

- In Table 1, the model trained on a noisy dataset outperforms the AnimateAnyone baseline trained on a clean TikTok dataset. Is this improvement due to the dataset or the proposed modules? If the improvement stems from the dataset, could the authors provide a more detailed analysis of dataset quality (e.g., pose diversity, data scale, character domain diversity)? If the improvement is due to the proposed modules, the authors should train their model solely on the TikTok dataset to directly compare with the baseline methods.

- In the inference stage, can the optical flow guider take a specific optical flow map as input to achieve results with camera rotation effects?

---

> ### Author Response · Authors · 2024-11-22
> **Response to Reviewer TMXE**
>
> **Q1**: Animation results of multiple characters (more than two), dense crowd, and high occlusion ratio.
>
> **A1**: We have updated the supplementary video, including the results of over two characters, dense crowd animations, and high occlusion ratios (0:03–1:02 in the video).  The results follow the conclusion that scenarios involving more people lead to poorer outcomes. The main reasons are as follows: 1) scenarios with more people present greater challenges, 2) videos containing a higher number of people are rare in the training data. We believe it is a point deserving future research and improvement.
>
> &nbsp;
>
> **Q2**: Lack of in-depth analysis related to multiple characters in the training dataset.
>
> **A2**: Thanks a lot for your insightful suggestion. We analyzed the distribution of character counts and body occlusion rates in our dataset. The corresponding charts and conclusions have been included in Appendix Section A.1 of the updated paper. Here, we briefly present the conclusion: 1) Single-character videos account for 66.25%, double-character videos account for 29.5%, and videos with triple-character and above account for a total of 4.25%. 2) 75% of videos have an average body occlusion rate of 0.21 or less, with 25% of videos having an occlusion rate of 0.05 or less.
>
> The proportions of videos with three or more characters and videos with high body occlusion rates in our dataset are relatively low, because the quantity of "videos with three or more characters" that can be collected is inherently very limited. With more data, we believe that our method could perform better in generating multi-character videos.
>
> &nbsp;
>
> **Q3**: Is the improvement from the dataset or the proposed modules?
>
> **A3**: Thanks for your valuable suggestions. We have completed this experimental result and included it in Table 1 (on page 6). The relevant analysis has also been updated in the updated paper (on page 9).
>
> Although the modules we designed primarily target challenges in noisy datasets and multi-character animations, our method still exhibits certain advantages when trained on the relatively clean Tiktok training set, particularly in terms of video evaluation metrics: FID-VID and FVD. However, the large-scale data collected in real-world scenarios is often noisy, and our main focus is on extracting useful information from this noisy data. We reproduce AnimateAnyone✝ on our noisy dataset but achieve worse results (in Table 1), indicating that our noisy dataset does not directly enhance the model's performance. However, when trained on the same noisy dataset, our method demonstrates significant improvement over AnimateAnyone✝, indicating that our improvement mainly comes from the designed modules.
>
> &nbsp;
>
> **Q4**: Can the method generate results with specific camera effects by taking specific optical flow as input?
>
> **A4**: Your brilliant idea holds great potential. Our method uses optical flow to decouple the background into static and dynamic components, effectively addressing the issue of unstable backgrounds. To achieve control over background motion, a potential improvement to our method is to replace the precise values of the background optical flow with their average values. Because the precise optical flow values are correlated with background elements, the absence of an effective method to predict precise optical flow during inference may lead to a misalignment between training and inference. Utilizing the mean optical flow is a potential approach to address this issue.
>
> We are excited about this idea and recognize that achieving this functionality requires further exploration. We will explore it as part of our future work.

---

> > ### Comment · Reviewer_TMXE · 2024-11-28
> >
> > Thanks for the detailed replies. The author(s) have addressed my question about the reason of model improvements and clarified the static information of their dataset. I would like to increase my score.

---

> > > ### Author Response · Authors · 2024-11-30
> > >
> > > Dear Reviewer TMXE,
> > >
> > > We would like to extend our heartfelt thanks for your thorough review of our paper. Your constructive suggestions and valuable comments have been pivotal in enhancing the quality of our work. We are committed to incorporating the new results and information, along with the associated discussions, as you suggested.
> > >
> > > Once again, we truly appreciate the time and effort you have dedicated to our paper.
> > >
> > > Best regards,
> > >
> > > The Authors

---

### Official Review · Reviewer_7Np5 · 2024-11-02

**Soundness:** 3
**Presentation:** 3
**Contribution:** 3
**Rating:** 6
**Confidence:** 3

**Summary:**

The paper proposes a new image-based pose retargeting/character animation diffusion model. The key contributions lie in how the proposed method handles (1) noisy backgrounds, which may degrade the quality due to spurious correlations, (2) explicitly capture the depth-order to enable more accurate generation for multi-character retargeting, and (3) enhance decoupling of appearances and body pose representation to achieve better texture quality.

The experimental results show that the proposed method achieves higher perceptual quality, both quantitatively and qualitatively.

**Strengths:**

The core strength of the method lies in the attention to the details. Specifically,
- By explicitly considering the depth order, the method disambiguates the spatial arrangement in multi-character settings, and therefore enables the model to learn better, and render with higher fidelity.
- By providing the pose for the reference image, the model can better decouple the texture and pose information (although it is still implicit). Despite being simple, this design appears to have a great positive impact on the generation quality (in Figure 10).
- The background optical flow mitigates the spurious correlations that hinder the model from focusing on character animation.

Overall, the presented method is sound, and the effectiveness of each proposed component is sufficiently verified through their experiments.

**Weaknesses:**

- The proposed method is computationally heavy (line 417-420, needs 8 A100 GPUs for training), making it less accessible to possible end users. This is, however, a common issue among diffusion-based methods, not particular to the paper.
- The depth-ambiguity is not completely resolved. One can see in the supplementary video 00:44, the top-left character has their hand rendered to the back incorrectly.
- The method focuses on front-facing poses. It is unclear how it performs on extreme poses with 360-degree rotation, and also poses like handstand, etc.

**Questions:**

I am overall positive about the submission. It would be nice if the authors can discuss a bit more about the following questions, in addition to the weaknesses raised above:
- What is the inference cost? Would it be able to run on commodity-level GPUs (e.g., Nvidia 2060, 3070, etc). I may have missed it, but it seems like only A100 is mentioned, and that is for training.
- How does the ```Ref Pose Guider``` handle multi-character skeleton maps. Do we assume there is a fixed number of character in the video?

---

> ### Author Response · Authors · 2024-11-22
> **Response to Reviewer 7Np5**
>
> **Q1**: Training and inference cost.
>
> **A1**: As you have pointed out, the high resource consumption during training is a common problem for diffusion-based methods. Our method aligns with other methods in terms of training resources: Champ and AnimateDance use 8 A100 GPUs, while AnimateAnyone uses 4 A100 GPUs. The inference resource requirements of our method are correlated with the resolution of the generated video. Specifically, generating 640\*896 videos requires 24GB of VRAM, 480P (480\*854) videos requires 16GB of VRAM, 360P (360\*640) videos requires 12GB of VRAM. Thus, our model can run on most commodity-level GPUs with the necessary adjustment to an appropriate video generation resolution. We have included the relevant information in Appendix Section C.1 of the updated paper.
>
> &nbsp;
>
> **Q2**: The issue of depth ambiguity.
>
> **A2**: Thanks for your careful watching. We primarily address the prominent challenge in multi-character animation: the ambiguity of occluded body parts. The mentioned case is about single-character animation. The depth order in our method is specifically for distinguishing multiple characters. In fact, single-character animations do not exhibit depth-ambiguity issues. This pointed-out particular case occurs simply because of the overly complex patterns on the clothing, because all the other cases on the same screen are animated normally, indicating it is not caused by "depth ambiguity".
>
> &nbsp;
>
> **Q3**: Animation results of extreme poses.
>
> **A3**: We have updated the supplementary video, including the results of 360-degree rotation (1:02–1:12 in the video). Our method can generate character animations of non-frontal and rotational poses, because the different limbs in the skeleton map are represented by distinct colors, which provide rich pose information, including rotation.
>
> &nbsp;
>
> **Q4**: Does this paper assume fixed number of characters?
>
> **A4**: Ref Pose Guider directly encodes the multi-character skeleton maps and inputs it into the network. We do not need to preset the number of characters. The network generates animations for the characters in the reference image based on the number and positions of characters in the multi-character pose sequence.

---

> > ### Comment · Reviewer_7Np5 · 2024-11-24
> >
> > Thanks for the detailed reply. Most of the concerns I have are addressed.
> >
> > I would like to comment a bit more on the depth-ambiguity issue:  The rebuttal states that "single-character animations do not exhibit depth-ambiguity issues". I am not convinced this is true.
> >
> > Since the pose extractor appears to be 2D-based (correct me if I am wrong), the model would naturally have depth ambiguity issues as the body parts' (relative) depths are not explicitly used for conditioning. I agree that the artifact in 00:44 could result from complex cloth patterns. That said, 00:50-00:53 (top row) in the original supp video shows clear artifacts of depth ambiguities, where the model fails to generate consistent hand orders (or even fails to generate at all).

---

> > > ### Author Response · Authors · 2024-11-28
> > > **Response to Reviewer 7Np5**
> > >
> > > Thank you to the reviewer for correcting our inaccurate expression in the rebuttal. Our original intention of stating that "single-character animations do not exhibit depth-ambiguity issues" is to express that the Depth Order Condition proposed in our paper will not affect the dance performance of a single person, because the Depth Order Condition is mainly used to distinguish the depth relationship of different people in multi-person scenes.
> > >
> > > At the same time, we agree with the reviewer's definition of "depth ambiguity issues" for a single person, and believe this is a very interesting problem for future exploration. For example, when the character's arm skeleton and body skeleton overlap, the arm skeleton may be in front of the body skeleton. One potential preliminary solution is to add a channel (which can be understood as an alpha channel) to the input RGB skeleton condition. i.e., the skeleton with a smaller depth has a larger alpha channel (e.g., the arm), and vice versa (e.g., the body). This is a problem worth exploring in the future.

---

> > > ### Author Response · Authors · 2024-11-30
> > >
> > > Dear Reviewer 7Np5,
> > >
> > > We would like to extend our appreciation for your time and valuable comments.
> > >
> > > We are eagerly looking forward to receiving your further valuable feedback and comments on the points we addressed in the response. Ensuring that the rebuttal aligns with your suggestions is of utmost importance.
> > >
> > > Best regards,
> > >
> > > The Authors

---

### Official Review · Reviewer_sEw7 · 2024-11-03

**Soundness:** 2
**Presentation:** 3
**Contribution:** 3
**Rating:** 5
**Confidence:** 5

**Summary:**

In this work, the authors propose solutions for challenges in image animation, such as instability under complex backgrounds and multi-person scenarios. To enhance quality in these situations, three additional guides are introduced: optical flow, depth order, and reference pose map. The network design is similar to previous methods which include ReferenceNet, denoising U-Net, and inserting a temporal module in the second stage. Experiments demonstrate that the proposed methods outperform existing approaches, but more experimental results should be provided.

**Strengths:**

1. This paper proposes to integrate three additional guiders, optical flow, depth order, and reference pose map to improve the animation performance under complex backgrounds and multi-character situations (mostly two persons).

2. The animation results are impressive and better than other methods, which demonstrate the efficiency of the proposed solutions.

**Weaknesses:**

1. The claim that the depth order map provides positional information for multiple characters is somewhat confusing. In practice, you first use the dilate-skeleton map to determine the positions of individuals, and subsequently obtain the depth order using depth information.

2. The results in the paper only involve two individuals. To support the claim of handling multiple characters, it would be beneficial to include examples with more than two characters.

3. The pose sequences for two persons in a video are all the same, I think more results in different pose sequences for different characters should be provided.

4. The network design follows the previous methods, like MagicAnimate, which includes ReferenceNet, denoising U-Net, and inserting a temporal module in the second stage. I think Figure 3 can be improved to better demonstrate the network architecture.

**Questions:**

1. I am curious about scenarios where two individuals have significant interactions.

2. I think that the results in Figure 2 should reference the specific previous methods used for these issues.

---

> ### Author Response · Authors · 2024-11-22
> **Response to Reviewer sEw7**
>
> **Q1**: Confusing claim of "depth order map provides positional information".
>
> **A1**: This confusion is caused by our writing. The depth guider provides order information of multiple characters in front of the camera, which guides the model to generate accurate relative positions of the characters. Specifically, when the limb skeletons of multiple characters intersect in a region, the model can correctly generate the limbs of the character in the foreground, rather than those behind. We referred to this as "positional information" in the original writing. The "positional information" does not mean spatial information but the order information. Thanks for pointing out this. We have been aware of the confusion here, and have provided a more precise description (on page 5) in the revised version.
>
> &nbsp;
>
> **Q2**: Animation results of multiple individuals.
>
> **A2**: Please refer to the general response for all reviewers and review our updated supplement again for the visual results (0:51–1:02 in the video).
>
> &nbsp;
>
> **Q3**: Animation results of different pose sequences.
>
> **A3**: Thanks for the comment. In our original version, we have included results for different sequences. In fact, our method is not specified to special pose sequences. It is universal for different sequences provided. Please refer to the general response for all reviewers and review our updated supplement again for the visual results (0:03–0:51 in the video).
>
> &nbsp;
>
> **Q4**: Figure 3 can be improved for better illustration.
>
> **A4**: Thanks a lot for your suggestion. It is indeed insightful. Per your suggestion, we have revised Figure 3 (on page 3) in the updated version of the paper. Please review the updated version for your reference.
>
> &nbsp;
>
> **Q5**: Animation results in the scenery of significant interaction.
>
> **A5**: Thanks for pointing it out. We have updated the supplementary video, including the results of multiple characters (over two), different pose sequences for different characters, and significant interactions between two individuals (0:03–1:02 in the video). Regarding the specifically concerned results of significant interactions, our method can adequately handle cases of severe occlusion among multiple characters.
>
> &nbsp;
>
> **Q6**: Results in Figure 2 should refer the previous methods.
>
> **A6**: Thanks for your suggestion, which will make our representation more accessible. We have referred to the specific methods in Figure 2 (on page 2) in the updated version.

---

> > ### Comment · Reviewer_sEw7 · 2024-11-24
> >
> > Hi authors, thanks for the efforts and response. However, I think the results shown in the new supplementary video are not convincing:
> >
> > 1. At timestamps 00:07 (left group) and 00:26 (right group), the method fails to preserve facial identities, which is crucial for image animation.
> > 2. The results at 00:30 and 01:20 (left group) appear too shaky, indicating instability in the method's performance.
> > 3. Relying on depth information for order handling seems insufficiently robust. This is evident in the occlusion issues observed at 00:55 (both groups), where the method does not effectively manage overlapping elements.
> >
> > Overall, while the topic of your research is promising, the results presented do not sufficiently verify the effectiveness of the proposed methods. Consequently, I will maintain my original score.

---

> > > ### Author Response · Authors · 2024-11-28
> > > **Response to Reviewer sEw7**
> > >
> > > Thank you for the reviewer's reply. We hope to clarify the three issues mentioned by the reviewer in the rebuttal stage so that the reviewer can better understand the superiority and effectiveness of our proposed method.
> > >
> > > 1. Facial problems
> > >
> > > We agree with the reviewer that face reconstruction is important for motion animation. Technically speaking, the face cases in animation tasks can be divided into frontal faces and complex facial poses (back face, side face, etc.)
> > >
> > > A. Frontal faces
> > >
> > > This is the most common case and is widely used in academia and industrial products. Both previous works (e.g., AnimateAnyone) and public benchmarks (e.g., TikTok, TED datasets) mainly focus on frontal face motion animation. We emphasize that our method is significantly better than existing methods in this scenario. In the original_supplement_video.mp4 of the supplementary materials, it can be found that the face reconstruction of most characters is satisfactory. Furthermore, we compare with SOTA methods (e.g., Moore-AnimateAnyone, MagicPose) in the newly submitted video newest_supplement_video_for_rebuttal.mp4 (00:01-00:17) of supplementary materials to further highlight the superiority of our method. In addition, the results of our method on the public datasets TikTok (Table 1 in the paper) and TED (Table 2 in the paper) also show that our method is better than existing methods in full motion animation (including face).
> > >
> > > B. Complex facial poses (back face, side face, etc.)
> > >
> > > This is an interesting problem and can be studied as an isolated topic in the motion animation community. Although our paper does not focus on this problem, we find that our method still outperforms previous methods in this scenario, please see the new video newest_supplement_video_for_rebuttal.mp4 (00:18-00:49) of supplementary materials. In the future, we will further explore this common problem in motion animation community.
> > >
> > > 2. Character shaking
> > >
> > > This problem is mainly caused by the pose detector (dwpose) rather than the shortcoming of our method.
> > > We notice that the dwpose output is a very shaky skeleton sequence for the original template video of the video mentioned by the reviewer.
> > > We recheck all skeletons used in the first submitted video (i.e., original_supplement_video.mp4) and found that if the extracted skeleton sequence is smooth, the characters generated by our method will not have this shaky problem.
> > > To demonstrate this problem, we have updated this visualization in new submitted video newest_supplement_video_for_rebuttal.mp4 (00:51-01:09).
> > > Addition, the animation videos used by smooth skeleton sequence (01:11-01:26) are also provided for better comparison.
> > > We speculate that this is a natural defect of dwpose when processing videos since it is an image-based method.
> > > However, considering that most previous methods (e.g., AnimateAnyone) use this tool for the pose detector, our paper follows their setting for fair comparison.
> > >
> > > 3. Overlapping problem
> > >
> > > Our method can effectively handle occlusion problems in multi-person scenarios. The analysis is as follows:
> > >
> > > A. Two character
> > >
> > > Our method performs well on occlusion problems in this scenario. In order to emphasize the advantages and effectiveness of our method, we show the results of previous methods (e.g., Moore-AnimateAnyone, MagicPose) in the newly submitted video newest_supplement_video_for_rebuttal.mp4 (01:27-01:36) for comparison, and it can be found that they cannot handle occlusion problems
> > >
> > > B. Three or more character
> > >
> > > Our method has a few flaws in this scenario. We believe that this is mainly due to the small proportion of such data in the dataset. As shown in Fig. 12 of our paper, the animation data of three or more people only accounts for (4.25%). Despite this, the performance of our method is much better than the previous method. The comparison is added in the newly submitted video newest_supplement_video_for_rebuttal.mp4 (01:36-01:46), which proves the ability of our method to handle overlapping problems.
> > >
> > > Conclusion
> > >
> > > Finally, we would like to further demonstrate the effectiveness of our method from the following aspects.
> > >
> > > A. We have newly provided ablation experiment comparison videos (i.e., newest_supplement_video_for_rebuttal.mp4 (01:48-02:10)) to illustrate the effectiveness of the three modules (i,e., Optical Flow Guider, Depth Order Guider and Ref Pose Guider) we proposed, and the ablation experiment results on the full testset of TikTok Dataset (Table 4 in the paper) also support this.
> > >
> > > B. The results in Table 1, Table 2 and Table 3 of the paper show that our proposed method is significantly better than the existing methods in both single-person and multi-person motion animations. These comprehensive and objective evaluation results prove the superiority and effectiveness of our method.

---

> > > > ### Author Response · Authors · 2024-11-29
> > > > **Kind reminder**
> > > >
> > > > Dear Reviewer,
> > > >
> > > > Have you had a chance to look at our rebuttal and the new additional submission video (newest_supplement_video_for_rebuttal.mp4)? We're eagerly awaiting your response to better address your concerns.
> > > >
> > > > Thank you very much.
> > > >
> > > > Best regards
> > > >
> > > > Authors

---

> > > > ### Author Response · Authors · 2024-12-03
> > > >
> > > > Dear Reviewer sEw7,
> > > >
> > > > Thank you very much for the time and effort in reviewing our work. We hope that our latest response has addressed your concerns. As the discussion is closing soon, please let us know if you have further questions.
> > > >
> > > > Once again, we sincerely appreciate your insightful and constructive comments.
> > > >
> > > > Best regards,
> > > >
> > > > The authors

---

### Author Response · Authors · 2024-11-22
**General Response to All Reviewers**

We would like to thank all the reviewers for their constructive comments. In general, reviewers acknowledge that, "the results are impressive and better than existing methods" (Reviewer sEw7), "the paper is well motivated and the proposed components are effective and sufficiently verified by experiments" (Reviewer 7Np5), "generates high-quality character animations when trained on noisy data (Reviewer TMXE), and "the skeletal dilation is novel", "the proposed methods are well-motivated", "it shows better quality on multi-character animation compared to baselines" (Reviewer eUry). Our method aims to solve the challenges concerning the stability in complex background and the tasks involving the multi-character animation, introducing multiple well-designed conditions, to handle the stable background, occlusion, and the decoupling of texture and pose.

&nbsp;

We also find there are the following concerns shared by reviewers, 1) missing results of more than two characters, 2) missing results of more motion sequences, 3) missing results in more difficult cases like extreme poses, heavy occlusion, or significant interactions. To this end, we have updated the supplementary video (new_supplement_video_for_rebuttal.mp4) to include these results of interest to the reviewers, while also incorporating them into Appendix Section E.3 of the updated paper. Please kindly review it again.

&nbsp;

In the following, we will respond to the comments from the individual reviewers separately. To ease the reviewing of the paper, we highlight the updated content with red color in the updated version. Please also be sure to review it again.

---

> ### Author Response · Authors · 2024-11-28
> **General Response to All Reviewers (new video added in the supplementary material)**
>
> To better demonstrate the superiority and effectiveness of our method, we have uploaded a new video (newest_supplement_video_for_rebuttal.mp4) in the supplementary material. Thus our supplementary materials contain 3 videos.
>
> 1. original_supplement_video.mp4
>
> Original submission video
>
> 2. new_supplement_video_for_rebuttal.mp4
>
> The first video submitted during the rebuttal phase, this video focuses on more multi-character and single-character results.
>
> 3. newest_supplement_video_for_rebuttal.mp4
>
> The second video submitted during the rebuttal phase, this video mainly focuses on the comparison results with other methods.

---

### Meta-Review · Area_Chair_bdJU · 2024-12-22

**Metareview:**

The paper tackles challenges in character image animation with a multi-condition framework using optical flow, depth order, and reference pose guiders to handle complex backgrounds and multiple characters. It outperforms existing methods in quality and introduces a new benchmark for standardized evaluation.

Strengths:
Innovative Design: Introduces optical flow, depth order, and reference pose guiders to improve animation in complex and multi-character scenes.
High Quality: Outperforms baselines with better results in multi-character and noisy background scenarios.
Enhanced Accuracy: Depth-order and reference pose maps improve spatial precision and texture-pose decoupling.

Weaknesses:
Depth and Multi-Character Limitations: The depth order map’s claim of providing positional information is unclear, and the method's evaluation is limited to two characters, lacking examples with more diverse multi-character scenarios or varying pose sequences.
Interaction and Pose Diversity: The method focuses on front-facing poses with limited results on interactions or extreme poses (e.g., 360-degree rotation, handstands), and facial regions are underrepresented, leaving questions about animation quality in diverse scenarios.
Clothing Animation and Skeletal Dilation Map: The skeletal dilation map restricts animation of loose clothing, with insufficient analysis of how dilation kernel size impacts performance.

We have decided to (weakly) accept the paper based on the mixed reviews, but we encourage the authors to improve both the manuscript and the supplementary video. Additionally, it is noted that there is significant room for improvement in the quality of the video results.

**Additional Comments On Reviewer Discussion:**

The authors addressed key concerns with updates, but reviewers had mixed reactions:

Facial Identity and Stability (Reviewer sEw7): Noted issues with facial identity preservation, stability, and occlusion in the supplementary video. The concerns were not resolved, and the reviewer maintained their negative score.
Multi-Character and Dataset Analysis (Reviewer TMXE): Added examples of more than two characters and dataset analysis, satisfying the reviewer, who increased their score from negative to positive.

While some concerns were addressed effectively, unresolved issues in the video results highlight areas needing improvement.

Based on these mixed reviews, we accept this borderline paper, because it is the SOTA although needing much improvement.

---

### Decision · Program_Chairs · 2025-01-22

Accept (Poster)